# mTOR-mediated cancer drug resistance suppresses autophagy and generates a druggable metabolic vulnerability

Niklas Gremke[1], Pierfrancesco Polo[1], Aaron Dort[1], Jean Schneikert[1], Sabrina Elmshäuser[1], Corinna Brehm[2], Ursula Klingmüller[3,4], Anna Schmitt[5], Hans Christian Reinhardt[5], Oleg Timofeev [1], Michael Wanzel[1,6,8] & Thorsten Stiewe [1,6,7,8]✉

Cancer cells have a characteristic metabolism, mostly caused by alterations in signal transduction networks rather than mutations in metabolic enzymes. For metabolic drugs to be cancer-selective, signaling alterations need to be identified that confer a druggable vulnerability. Here, we demonstrate that many tumor cells with an acquired cancer drug resistance exhibit increased sensitivity to mechanistically distinct inhibitors of cancer metabolism. We demonstrate that this metabolic vulnerability is driven by mTORC1, which promotes resistance to chemotherapy and targeted cancer drugs, but simultaneously suppresses autophagy. We show that autophagy is essential for tumor cells to cope with therapeutic perturbation of metabolism and that mTORC1-mediated suppression of autophagy is required and sufficient for generating a metabolic vulnerability leading to energy crisis and apoptosis. Our study links mTOR-induced cancer drug resistance to autophagy defects as a cause of a metabolic liability and opens a therapeutic window for the treatment of otherwise therapy-refractory tumor patients.

[1] Institute of Molecular Oncology, Philipps-University, Marburg, Germany. [2] Institute of Pathology, Philipps-University, Marburg, Germany. [3] Division Systems Biology of Signal Transduction, German Cancer Research Center (DKFZ), Heidelberg, Germany. [4] Translational Lung Research Center Heidelberg (TLRC), German Center for Lung Research (DZL), Heidelberg, Germany. [5] Clinic for Hematology and Stem Cell Transplantation, West German Cancer Center, University Hospital Essen, German Cancer Consortium (DKTK), Essen, Germany. [6] Universities of Giessen and Marburg Lung Center, German Center for Lung Research (DZL), Marburg, Germany. [7] Genomics Core Facility, Philipps-University, Marburg, Germany. [8]These authors contributed equally: Michael Wanzel, Thorsten Stiewe. ✉email: stiewe@uni-marburg.de

Cancer therapy has improved significantly over the last decades, but ultimately still fails in almost half of all patients owing to the development of therapy resistance. The inherent genomic instability and phenotypic plasticity allow tumors to rapidly evolve and give rise to drug-resistant subclones as a cause of intrinsic or acquired therapy resistance and eventually lead to tumor relapse and therapy failure[1–3].

A key driver of cancer drug resistance is the mTOR (mechanistic target of rapamycin) kinase, which acts in two distinct mTOR complexes, mTORC1 and mTORC2, to integrate a diverse set of environmental cues, such as growth factor signals and nutritional status, to direct eukaryotic metabolism and cell growth[4–7]. In a growth-promoting microenvironment, mTOR switches cell metabolism to increased production of protein, lipids and nucleotides, while suppressing catabolic pathways. Most notably, mTOR inhibits the process of autophagy whereby intracellular components are engulfed in double-membraned vesicles, the autophagosomes, which ultimately fuse with lysosomes where the contents are degraded and recycled into the cytosol[8]. Autophagy is initiated under conditions of nutrient starvation and metabolic stress, for example by AMPK (AMP-activated protein kinase), which in response to a drop in cellular energy charge phosphorylates ULK1 (Unc-51 Like Autophagy Activating Kinase 1)[9]. mTORC1 also phosphorylates ULK1, but this phosphorylation is inhibitory, counteracts ULK1 activation by AMPK, and thereby prevents the initiation of autophagy under nutrient-replete conditions[9,10].

Although relatively few cancers harbor constitutively activating mutations in the mTOR gene or direct pathway regulators[11,12], both mTOR complexes are essential effectors of the most common oncogenic drivers, including those in the Ras-driven MAPK and PI3K–AKT pathways[7]. As such, sustained mTOR signaling causes resistance to therapeutics targeted against the driving oncogenes, for instance, in lung cancer, breast cancer, and melanoma[4,13–16]. In addition, mTOR is implicated in chemotherapy resistance, for example, by inducing the Fanconi anemia DNA repair pathway, which resolves cytotoxic DNA interstrand crosslinks generated by platinum compounds like cisplatin (CDDP)[17–20].

mTOR inhibitors are therefore considered a valuable addition to chemotherapy or targeted cancer therapy, either as an option for relapsed patients or as a frontline combination therapy to prevent or delay the development of resistance due to sustained mTOR signaling[4,5]. However, first-generation mTOR inhibitors, termed rapalogs, which prevent phosphorylation of only some mTORC1 targets, have shown little efficacy outside certain exceptional contexts and use of second-generation, ATP-competitive, catalytic mTORC1/mTORC2 inhibitors might be limited by more severe toxicities owing to their broad effects[7,11,21].

Given the prominent role of mTOR in cancer drug resistance and the yet limited clinical success with mTOR inhibitors, we aimed to identify novel therapeutic options for cancer cells with mTOR-dependent drug resistance. In light of the central role of mTOR as a rheostat of cell metabolism[22], we examined drugs targeting energy metabolism, which is reprogrammed in cancer cells to produce the enormous amounts of biomass needed for sustained growth. Among other alterations, cancer cells commonly upregulate glycolysis even in the presence of oxygen, known as the "Warburg effect" or aerobic glycolysis. This characteristic metabolic phenotype is considered a hallmark of cancer and attributed to a special dependence of highly proliferating cells on biosynthetic pathways that use intermediates derived from glycolysis[23–25].

At the molecular level aerobic glycolysis is achieved, for example, by increased glucose uptake and inhibition of glycolysis downstream at the steps catalyzed by pyruvate kinase and pyruvate dehydrogenase. These processes can be pharmacologically targeted with 2-deoxy-D-glucose (2DG) and dichloroacetate (DCA), respectively[26]. Studies using 2DG and DCA have provided deeper mechanistic insight into cancer cell metabolism and have led to the development of novel therapeutic strategies to exploit altered metabolism[27]. 2DG and DCA have also been tested as anti-cancer agent in patients and showed relatively high toxicity, which limits their clinical use[28–31]. Several newer compounds are currently tested in clinical trials and many more are in preclinical development, but—given that the characteristics of cancer metabolism are mostly a consequence of alterations in signal transduction networks rather than direct mutations in metabolic enzymes—cancer-selectivity remains a problem so that many drugs are still limited more by toxicities than by their ability to kill cancer cells[27]. There is therefore an urgent need to identify signaling alterations in cancer cells that confer a druggable metabolic vulnerability sufficient to open a therapeutic window.

Here, we report that the constitutive activation of mTOR signaling in tumors with acquired resistance to chemotherapy and targeted therapeutics suppresses autophagy and thereby deprives cells of a powerful survival mechanism to compensate metabolic perturbation induced by various compounds targeting cancer metabolism. This identifies a druggable vulnerability of therapy-refractory cancer cells with mTOR-dependent drug resistance.

## Results

**Metabolic vulnerability of cisplatin-resistant lung cancer.** The dismal prognosis of cancer patients with acquired cancer drug resistance prompted us to explore vulnerabilities of chemotherapy-resistant cancer cells. We used H460 cells from a KRAS$^{Q61H}$-mutant, p53 wild-type non-small cell lung cancer (NSCLC) patient, adapted the parental H460 (H460$^{par}$) cells by dose escalation to CDDP, a standard component of first-line chemotherapy for NSCLC, and obtained CDDP-resistant H460$^{res}$ cells[19,32]. When examining the resistance pattern, H460$^{res}$ cells were not only resistant to CDDP, but also to many other DNA damaging chemotherapeutics including other platinum compounds, etoposide, and doxorubicin (Fig. 1a). In striking contrast, proliferation of H460$^{res}$ cells was severely inhibited by the metabolic inhibitors 2DG and DCA through induction of apoptosis, although both compounds had insignificant effects on parental cells (Fig. 1a–d). Both 2DG and DCA target glucose metabolism: 2DG competitively inhibits hexokinase to slow glucose uptake, whereas DCA targets the inhibitory pyruvate dehydrogenase kinase, thereby reducing lactate production from glycolysis and promoting oxidative metabolism of pyruvate in the mitochondria[24].

To validate these observations in a second NSCLC model, we experimentally adapted H1975 cells, which are derived from an EGFR$^{L858R,T790M}$ and TP53$^{R273H}$ mutant NSCLC, to escalating doses of CDDP (Fig. 1e). Compared with parental H1975 cells, CDDP-resistant H1975 cultures showed strongly diminished clonogenic growth and increased apoptosis when treated with 2DG or DCA (Fig. 1f, g). To validate these findings for tumor cells, which have relapsed in vivo after CDDP therapy, we used conditional Kras$^{LSL-G12D/+}$;Trp53$^{flox/flox}$ mice. Following inhalation of Cre adenovirus, these mice developed lung adenocarcinomas (Kras$^{G12D/+}$;Trp53$^{Δ/Δ}$) that initially responded to CDDP therapy but eventually relapsed (Fig. 1h)[33]. Primary tumor cell cultures were obtained from relapsed mice and, compared with tumor cells from CDDP-naive mice, showed significantly increased resistance to CDDP, but hypersensitivity to 2DG and DCA (Fig. 1h, i). Thus, in multiple different lung adenocarcinoma models with different driver oncogenes and p53 co-mutations

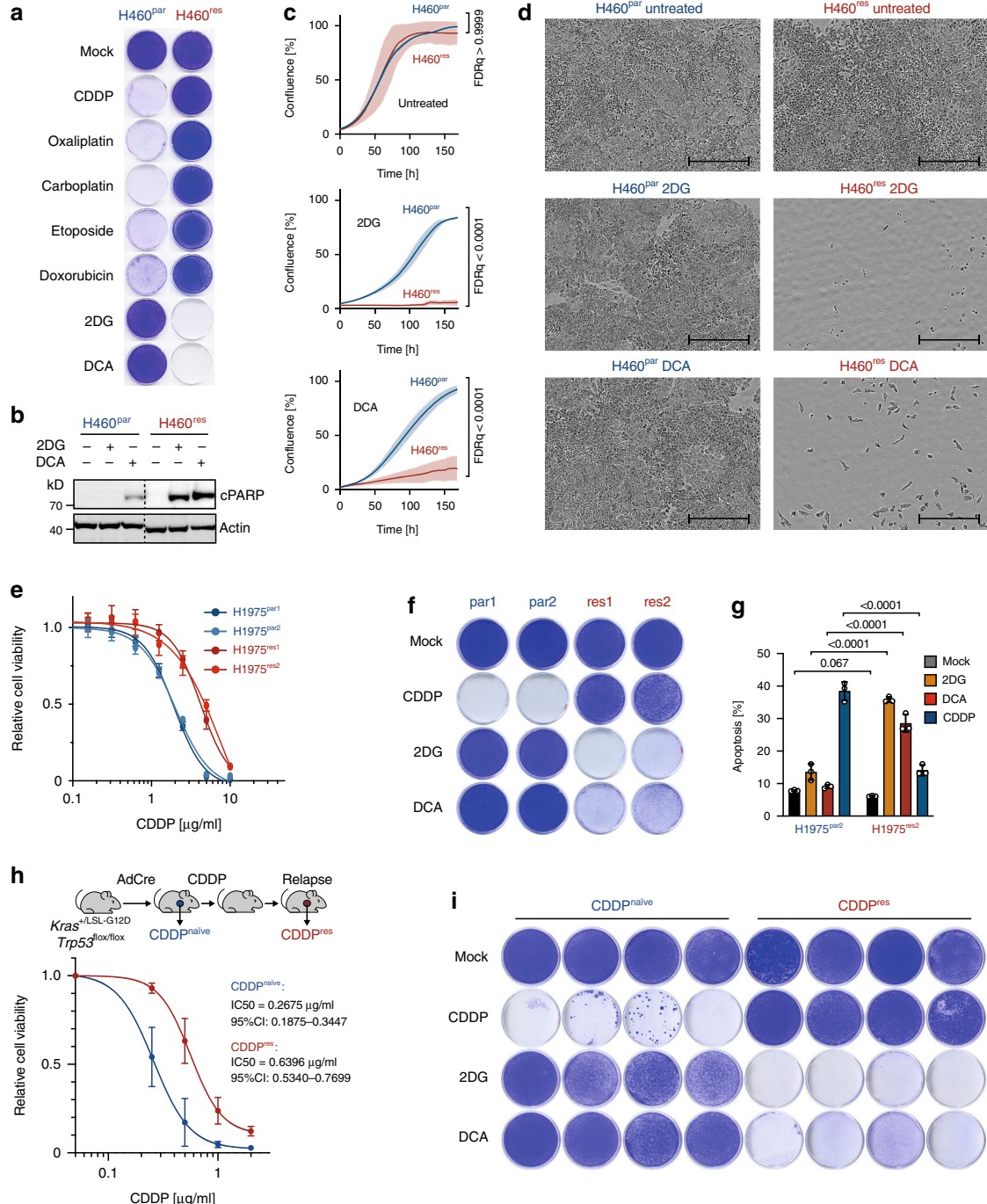

**Fig. 1 Metabolic vulnerability of cisplatin-resistant NSCLC. a** Clonogenic growth of parental H460[par] and CDDP-resistant H460[res] NSCLC cells treated with indicated drugs. **b** Western blot for cleaved Parp (cParp) as apoptosis marker. **c, d** Proliferation of untreated or 2DG/DCA-treated cells. **c** Cell culture confluency measured by real-time live-cell imaging. Shown are mean ± SD, $n = 3$, FDR $q$ values. **d** Representative images of cell cultures at the end of the treatment period. Scale bars, 400 µm. **e** Cell viability of two pairs of parental (par1, par2) and CDDP-resistant (res1, res2) H1975 cells treated for 3 days with CDDP. Shown are mean ± SD, $n = 3$. IC50 (95% CI): par1, 1.9 (1.7–2.0); par2, 1.9 (1.7–2.0); res1, 4.2 (3.9–4.7); res2, 4.5 (4.0–4.9). **f** Clonogenic growth of H1975 cells treated 5 days with indicated drugs. **g** Flow cytometry analysis for apoptosis (sub-G1) 4 days after treatment with 2DG, DCA, and CDDP. Shown are mean ± SD, $n = 3$, FDR $q$ values. **h** $Kras^{G12D/+};Trp53^{\Delta/\Delta}$ lung adenocarcinoma cells were obtained from either therapy-naive mice (CDDP[naïve]) or mice relapsed after in vivo CDDP therapy (CDDP[res]). Cell viability 72 hours following in vitro exposure to CDDP. Shown are mean ± SD, $n = 4$ cell lines, IC50 (95% CI). **i** Clonogenic growth of $Kras^{G12D/+};Trp53^{\Delta/\Delta}$ lung adenocarcinoma cells in the presence of indicated drugs. Shown are four independent CDDP[naïve] and CDDP[res] tumor cell cultures.

development of CDDP resistance generated a remarkable vulnerability to perturbation of glucose metabolism.

**Metabolic vulnerability is mediated by mTOR.** The exceptional 2DG/DCA hypersensitivity of various CDDP-resistant tumor cells suggested a mechanistic link between CDDP resistance and metabolic sensitivity and promised that delineating the underlying mechanisms would identify biomarkers for predicting metabolic drug responses. Hyperactive mTOR signaling is known to contribute to CDDP resistance, for example, by upregulating FancD2, an essential protein in the Fanconi anemia DNA repair pathway, which promotes repair of lethal CDDP-induced DNA interstrand crosslinks[17,19,34–36]. In line, CDDP-resistant H460, H1975 and mouse adenocarcinoma cells all displayed strongly elevated mTOR signaling, characterized by phosphorylation of mTORC1 target sites (p70S6K[T389], 4E-BP1[T37/46], and ULK1[S757]) and increased expression of FancD2 (Fig. 2a–c). Furthermore, the ATP-competitive mTOR kinase inhibitor AZD8055[37] and the mTORC1-selective inhibitors rapamycin and everolimus blocked phosphorylation of mTOR targets and effectively re-sensitized H460[res] cells to CDDP (Fig. 2a, d, Supplementary Figs. 1). Importantly, mTOR inhibition not only restored H460[res] cell sensitivity to CDDP but simultaneously blunted the apoptotic response to 2DG/DCA and rescued clonogenic growth in the presence of 2DG/DCA (Fig. 2d, e, Supplementary Fig. 1 and 2). Confirming the on-target specificity, siRNA-mediated mTOR knockdown as well as knockdown of the mTORC1 subunit Raptor protected H460[res] cells from 2DG and DCA, whereas knockdown of the mTORC2-specific subunit Rictor had no effect (Fig. 2f, g, Supplementary Fig. 1). In sum, pharmacological or RNAi-mediated inhibition of mTORC1 reverted the drug response pattern of CDDP-resistant tumor cells to that of parental cells (Fig. 2d). Together this indicates, that elevated mTORC1 signaling in CDDP-resistant tumor cells is not only required for the CDDP resistance phenotype but simultaneously essential for their metabolic vulnerability.

We next investigated whether upregulation of mTORC1 signaling can trigger 2DG/DCA hypersensitivity. When endogenous mTORC1 kinase activity was increased by enforced expression of the mTOR-activating GTPase Rheb1 (Ras homolog enriched in brain), H460[par] cells became sensitive to 2DG/DCA (Fig. 2h, i). Upon depletion of the Rheb1-inhibitory TSC subunits TSC1 and TSC2 mTORC1 signaling was further boosted, as evident from higher levels of phosphorylated p70S6K[T389], 4E-BP1[T37/46], and ULK1[S757], and resulted in even higher sensitivity to 2DG and DCA (Fig. 2h, i). Moreover, different hyperactive mTOR mutants, which were identified previously in cancer patients[12] and which induced robust mTORC1 signaling in H460[par] cells, sensitized to 2DG/DCA-induced apoptosis and led to impaired clonogenic growth in the presence of 2DG/DCA (Fig. 2j–l). We conclude that activation of mTORC1 signaling is not only required but also sufficient to induce metabolic vulnerability to 2DG and DCA.

**mTOR-mediated autophagy defect in CDDP-resistant tumor cells.** mTOR coordinates eukaryotic cell growth and metabolism with environmental inputs and regulates fundamental cell processes, from protein synthesis to autophagy[6]. Autophagy, in particular, is critical for cell survival under conditions of energy stress induced by nutrient starvation[38], suggesting that the autophagy-inhibitory function of mTORC1 might also be responsible for vulnerability to metabolic perturbation. Consistent with the known metabolic effects, 2DG/DCA treatment of H460[res] cells induced clear signs of energy stress from ATP-depletion, including phosphorylation of AMP-activated protein

kinase (AMPK) and the AMPK target site in acetyl-CoA carboxylase (ACC[S79]) (Fig. 2a).

Hypothesizing that the differential response of H460[par] and H460[res] cells to 2DG/DCA reflects mTOR-dependent differences in autophagic flux, we performed LC3 turnover assays[39]. To accurately quantify LC3 protein levels, LC3 was tagged with the short 11 amino-acid HiBiT peptide that is capable of producing bright and quantitative luminescence through high affinity complementation with NanoLuc[40]. As steady state levels of LC3 on their own do not address issues of autophagic flux[39], we measured LC3 accumulation after blocking autolysosomal degradation with chloroquine (Fig. 3a). LC3 accumulation in H460[res] cells was strongly reduced compared with H460[par] cells, indicating substantially lower autophagic flux already in the absence of 2DG/DCA. mTOR inhibition with AZD8055 restored LC3 accumulation in H460[res], with no discernible effect in H460[par] cells, indicating that the reduction of basal autophagic flux in H460[res] cells is mTOR-dependent.

Upon treatment with 2DG/DCA, LC3 levels dropped in a dose-dependent manner in H460[par] cells, suggesting induction of autophagy (Fig. 3b, c). Consistent with a 2DG/DCA-induced increase in autophagic flux, 2DG/DCA caused a dose-dependent accumulation of LC3 in chloroquine-treated H460[par] cells. Importantly, in H460[res] cells 2DG/DCA treatment had little to no effect on LC3 levels in the absence and presence of chloroquine, demonstrating that H460[res] cells fail to initiate autophagy in response to 2DG/DCA. However, this autophagy defect was completely rescued by mTOR inhibition with AZD8055.

The results from the LC3-HiBiT reporter assay were confirmed using cells expressing DsRed-LC3-GFP, which yields a diffuse cytoplasmic green fluorescence that, upon autophagy induction, shifts to red-fluorescent puncta marking autophagosomes[41]. Consistent with a defect of H460[res] cells to respond to 2DG/DCA treatment with autophagy, only H460[par] cells showed massive induction of red-fluorescent LC3 puncta under treatment (Fig. 3d, e).

Mechanistically, autophagy initiation is largely controlled in an antagonistic manner by mTORC1 and AMPK through phosphorylation of ULK1[9,10,42]. CDDP-resistant H460 and H1975 lung cancer cells showed increased levels of the mTORC1-dependent ULK1[S757] phosphorylation (Fig. 2a, b), which prevents ULK1 activation by AMPK[10]. Interestingly, ULK1[S757] phosphorylation was not only maintained, but increased even further under 2DG/DCA treatment, thereby providing a mechanistic explanation for the failure of 2DG/DCA to activate a protective autophagy response that could mitigate the energetic stress caused by metabolic perturbation (Fig. 2a).

**Suppression of autophagy by mTOR sensitizes to 2DG/DCA.** The previous results suggested that elevated and sustained mTOR signaling in H460[res] cells directly interferes with the 2DG/DCA-induced initiation of an autophagic survival response, which might protect H460[par] cells from 2DG/DCA toxicity. Supporting a protective role of autophagy in the context of metabolic perturbation, various pharmacological autophagy inhibitors, acting on different steps in the autophagy process, sensitized H460[par] cells to 2DG/DCA (Fig. 4a). Similarly, knockdown of the essential autophagy gene ATG7 rendered H460[par] cells susceptible to 2DG/DCA (Fig. 4b, c). Owing to the possibility of off-target effects and incomplete inhibition associated with pharmacological inhibitors and siRNAs, we also generated various H460[par] cell clones with CRISPR-induced ATG7 insertion/deletion-mutations (Fig. 4d–f). Only cell clones with complete ATG7 knockout showed accumulation of the autophagy cargo protein p62/SQSTM1 and failure of LC3-I to LC3-II processing as confirmation for their autophagy

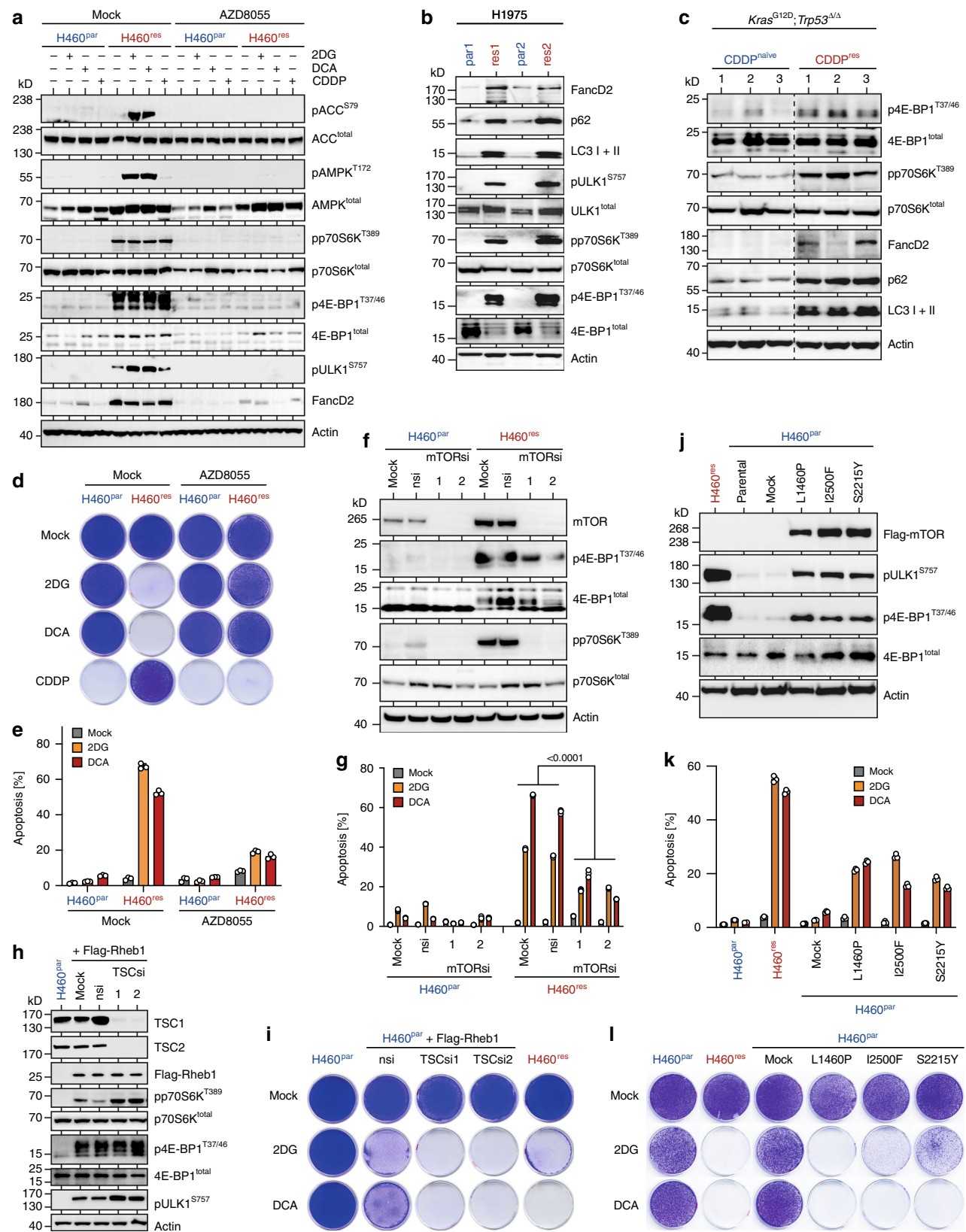

deficiency (Fig. 4d). Exactly these clones also showed impaired clonogenic growth and proliferation under 2DG/DCA treatment (Fig. 4e, f). Further validating a specific role of autophagy as opposed to non-canonical ATG7 functions in LC3-associated phagocytosis (LAP) and endocytosis, we observed sensitization to 2DG/DCA also upon knockdown or knockout of ATG14 and

FIP200, two genes considered specific for autophagy, whereas knockdown of RUBCN, involved in LAP but not classic autophagy, showed no effect (Supplementary Fig. 3)[43,44]. Of note, autophagy deficiency by knockout of ATG7, ATG14, or FIP200 did not alter CDDP sensitivity (Supplementary Fig. 3). Together, these experiments specifically link autophagy deficiency to the

**Fig. 2 Metabolic vulnerability of CDDP-resistant tumor cells is mediated by mTOR. a** Western blot of parental H460[par] and CDDP-resistant H460[res] cells treated for 48 hours as indicated. **b** Western blot of parental and CDDP-resistant subclones of the H1975 NSCLC cell line. **c** Western blot of $Kras^{G12D/+}$; $Trp53^{\Delta/\Delta\Delta}$ lung adenocarcinoma cells obtained from either therapy-naive mice (CDDP[naive]) or mice relapsed after CDDP therapy (CDDP[res]). Shown are three independent CDDP[naive] and CDDP[res] tumor cell cultures. **d, e** H460 cells were grown in the absence or presence of 2DG, DCA or CDDP ± AZD8055. **d** Clonogenic growth assay. **e** Flow cytometry analysis for apoptosis (sub-G1) 5 days after treatment. Shown are mean ± SD, $n = 3$, false discovery rate $q$ values (FDRq). **f, g** H460 cells were transfected with two independent mTOR siRNAs in comparison with mock transfection or non-targeting control siRNA (nsi). **f** Western blot. **g**, Apoptosis (sub-G1) flow cytometry analysis of H460 cells from **f** treated for 4 days with 2DG/DCA. Shown are mean ± SD, $n = 3$, two-way ANOVA. **h, i** H460[par] cells stably expressing Flag-Rheb1 were transfected with two independent sets of siRNAs targeting TSC. nsi, non-targeting control siRNA. **h** Western blot. **i** Clonogenic growth in the absence and presence of 2DG/DCA. H460[par] and H460[res] were included as control. **j-l**, H460[par] cells were stably transfected with indicated cancer-derived mTOR mutants or empty vector (mock). **j** Western blot. **k** Flow cytometry analysis for apoptosis (sub-G1) after 5 days treatment with 2DG/DCA. Shown are mean ± SD, $n = 3$. **l** Clonogenic growth in the absence and presence of 2DG/DCA. H460[par] and H460[res] were included as control.

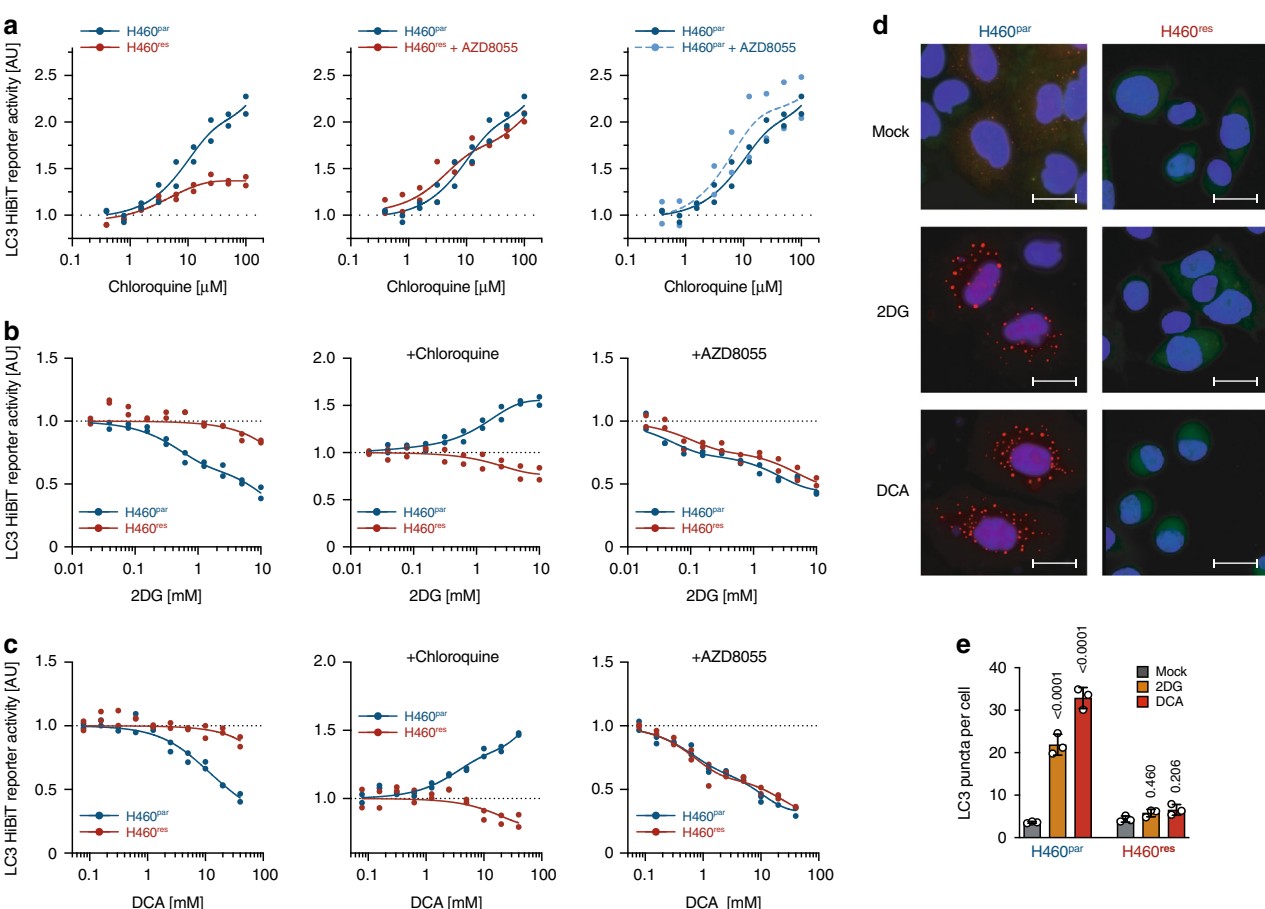

**Fig. 3 mTOR-mediated autophagy defect in CDDP-resistant tumor cells. a–c** Autophagic flux analysis of H460[par] and H460[res] cells stably transfected with LC3-HiBiT reporter. **a** Cells were pre-treated with AZD8055 for 48 hours before adding the indicated doses of chloroquine for 6 hours. **b, c** Cells were pre-treated as indicated with chloroquine or AZD8055 for 48 hours before adding increasing doses of **b** 2DG or **c** DCA for 6 hours. Shown is reporter activity measured as LC3-HiBiT luminescence normalized to untreated of $n = 2$ replicates. **d, e** H460 cells were transfected with DsRed-LC3-GFP, which yields a diffuse cytoplasmic green fluorescence that, upon autophagy induction, shifts to red-fluorescent puncta marking autophagosomes[25]. **d** Representative immunofluorescence images of DsRed-LC3-GFP-expressing cells following 48 hours treatment. Scale bars, 10 µm. **e** Quantification of **d**. Shown are mean ± SD, $n = 3$ independent experiments in each of which 100 cells were quantified, two-way ANOVA with Dunnet's multiple comparisons test.

antitumor activity of 2DG/DCA and highlight autophagy defects as a sufficient cause for metabolic drug vulnerability.

To examine whether the autophagy-inhibiting activity of mTORC1 is required for metabolic vulnerability, we expanded ATG7-modified H460[par] clones in the presence of escalating CDDP doses yielding CDDP-resistant H460[res] clones with different ATG7-mutation status. Independent of ATG7-status, all these CDDP-resistant clones were hypersensitive to 2DG/DCA, but only

ATG7-proficient clones were rescued from 2DG/DCA cytotoxicity by mTOR inhibition (Fig. 4g, h). This proves that the protection provided by mTOR inhibition is dependent on an intact autophagy pathway and, conversely, that mTORC1 is sensitizing to 2DG/DCA by suppressing autophagy. We conclude that autophagy ensures survival under metabolic perturbation stress and that upregulated mTORC1 signaling in CDDP-resistant tumor cells sensitizes to 2DG/DCA by interfering with this survival mechanism.

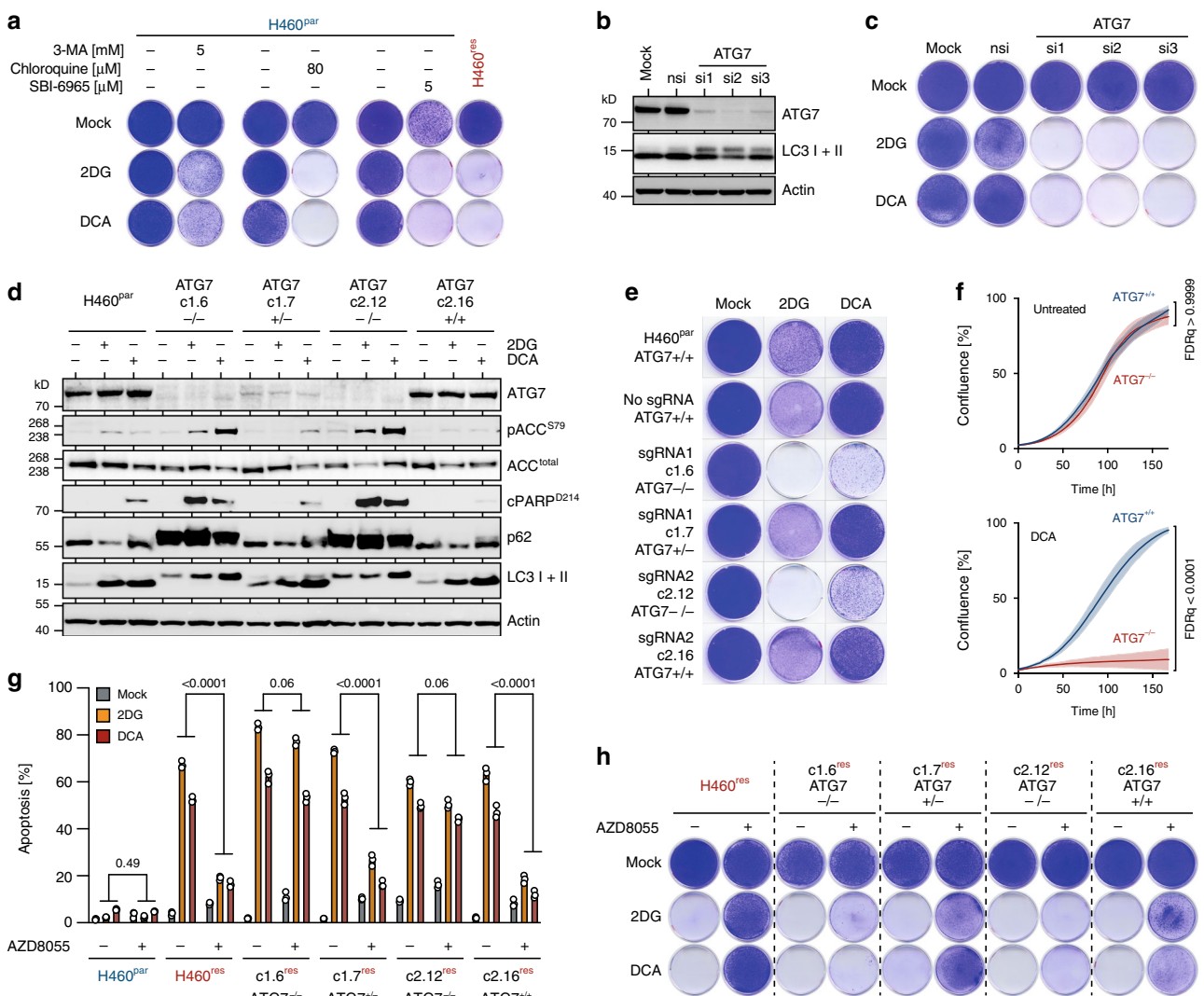

**Fig. 4 Suppression of autophagy by mTOR sensitizes to 2DG/DCA. a** Clonogenic growth assay. H460[par] cells were pre-treated with 3-methyladenine (3-MA), chloroquine or ULK inhibitor SBI-0206965 for 48 hours before adding 2DG/DCA. H460[res] cells are shown as control. **b, c** H460[par] cells were transfected with a control siRNA (nsi) or siRNAs targeting ATG7. **b** Western blot. **c** Clonogenic growth under 2DG/DCA treatment. **d–f** H460[par] subclones with indicated CRISPR-engineered ATG7 mutations were analyzed for mTOR signaling, autophagy markers and response to 2DG/DCA treatment. **d**, Western blot. **e**, Clonogenic growth. **f**, Proliferation curves determined by real-time live-cell imaging. Shown is cell culture confluence as mean ± SD, n = 3, FDR q values. **g, h** H460[par] clones with CRISPR-induced ATG7 indel mutations were made CDDP-resistant by dose escalation and tested for mTOR-dependent response to 2DG/DCA treatment. **g**, Flow cytometry analysis for apoptosis (sub-G1). Shown are mean ± SD, n = 3, FDR q values. **h** Clonogenic growth of CDDP-resistant H460 cells with indicated ATG7 genotype treated with 2DG/DCA ± AZD8055. Shown are representative images.

**mTOR-mediated metabolic vulnerability extends to biguanides.** By degrading and recycling intracellular content, autophagy can supply cells with a broad variety of metabolites. We therefore suspected that autophagy enables bypass of different metabolic blocks, so that suppression of autophagy would not only sensitize to 2DG/DCA but also to other metabolically active compounds. As DCA blocks phosphorylation of PDH E1 subunit α, we first tested the more selective PDH kinase inhibitor AZD7545[45]. DCA and AZD7545 both prevented PDH phosphorylation in H460[par] and H460[res] cells, but activated AMPK, induced cell death and inhibited clonogenic growth only in H460[res] cells (Supplementary Fig. 4a, b), thereby validating PDH kinases as a therapeutic target in tumor cells with mTOR-mediated therapy resistance.

As an entirely different class of metabolic compounds, we tested anti-diabetic biguanides metformin (Met) and phenformin (Phen), which exert pleiotropic effects on cancer metabolism by

inhibiting the mitochondrial electron transport chain complex I[46–48]. Similar as seen with 2DG/DCA, H460[par] cells reacted to Met/Phen with induction of autophagic flux as evidenced by dosage-dependent LC3 degradation in the absence and LC3 accumulation in the presence of chloroquine (Fig. 5a). In contrast, H460[res] cells demonstrated only negligible fluctuations in LC3 levels consistent with a failure of Met/Phen to induce autophagy (Fig. 5a). Instead, H460[res] cells displayed increased AMPK[T172] and ACC[S79] phosphorylation and PARP cleavage as signs of energy stress and ensuing apoptosis, respectively (Fig. 5b, left panel), and inhibition of clonogenic growth by Met/Phen (Fig. 5d).

Importantly, the differential response to Met/Phen treatment was also linked to mTORC1. mTOR inhibition restored LC3 degradation in Met/Phen-treated H460[res] cells (Fig. 5a), indicating efficient autophagy induction, and rescued these cells from energy stress and apoptosis (Fig. 5b, right panel, Supplementary

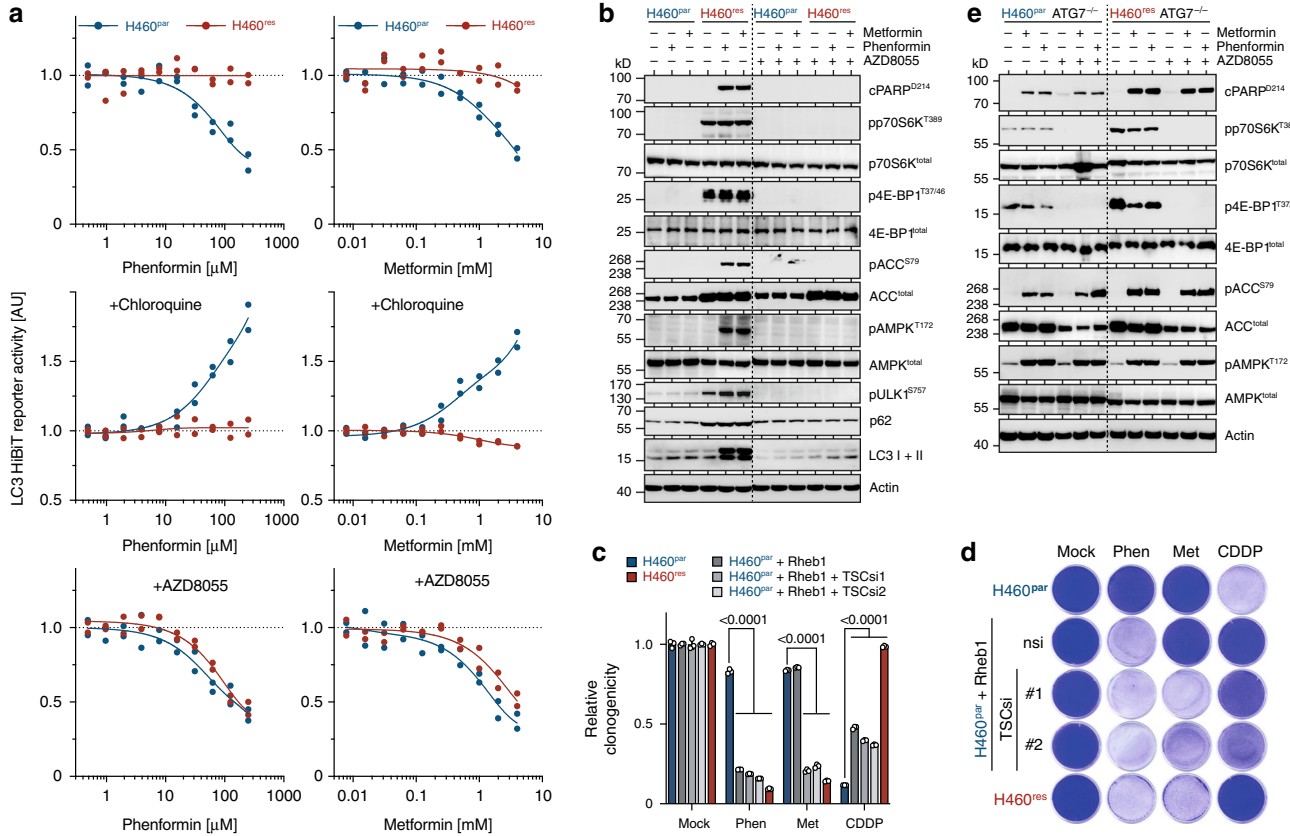

**Fig. 5 mTOR-mediated metabolic vulnerability extends to biguanides. a** Autophagic flux analysis. LC3-HiBiT-expressing H460[par] or H460[res] cells were pre-treated as indicated with chloroquine and AZD8055 for 48 hours and then treated with increasing doses of Metformin (Met) or Phenformin (Phen) for 6 hours. Shown is LC3-HiBiT reporter activity measured as luminescence normalized to untreated of $n = 2$ replicates. **b** Western blot. H460 cells were pre-treated with AZD8055 for 72 hours as indicated before adding Met/Phen for 6 hours. **c, d** Clonogenic growth of H460 cells expressing Flag-Rheb1 and subsequently transfected with siRNAs targeting TSC were treated with Met, Phen or CDDP. **c** Quantification shown as mean ± SD, $n = 3$, two-way ANOVA with Tukey's multiple comparisons test. **d** Representative images. **e** CDDP-sensitive, autophagy-deficient H460[par] ATG7[-/-] cells (clone c2.12 from Fig. 4) were made CDDP-resistant by dose escalation yielding H460[res] ATG7[-/-] cells. Western blot of cells treated with Met/Phen ± AZD8055 for 3 days.

Fig. 1). Furthermore, mTORC1 signaling enforced by Rheb1 expression and TSC depletion strongly reduced clonogenic growth of H460[par] cells treated with Met/Phen (Fig. 5c, d). Elevated mTORC1 signaling is therefore required and sufficient for Met/Phen sensitivity.

Likewise, an mTORC1-independent autophagy defect induced by ATG7, ATG14, or FIP200-deficiency caused cells to run into energy stress, triggered apoptosis and reduced clonogenic growth when treated with Met/Phen (Fig. 5e, left panel, Supplementary Fig. 3). Moreover, unlike CDDP-resistant H460[res] cells with an mTORC1-dependent autophagy defect (Fig. 5b), CDDP-resistant H460[res] cells that cannot induce autophagy because of an ATG7 knockout (H460[res]ATG7[-/-]) could not be rescued by mTOR inhibition (Fig. 5e, right panel), confirming that autophagy suppression by mTORC1 is causal for Met/Phen sensitivity. Of note, on their own ineffective low doses of Met/Phen and 2DG/DCA effectively suppressed clonogenic growth when combined, without losing specificity for mTOR-activated (H460[res]) cells, suggesting that mechanistically distinct metabolic drugs can be combined at more tolerable low doses to effectively eradicate mTOR-activated tumor cells while sparing normal cells (Supplementary Fig. 4c).

**Metabolic vulnerability linked to targeted drug resistance.** mTOR is also a frequent cause for resistance to targeted therapies[4,5,49]. For example, we recently demonstrated mTOR-mediated resistance to the experimental p53-activating drug

RITA[19,50]. Cell lines from multiple different tumor entities with acquired resistance to RITA displayed upregulated mTORC1 signaling accompanied by defective autophagy and hypersensitivity to 2DG/DCA (Supplementary Fig. 5). This prompted us to test for metabolic vulnerability in tumors with mTORC1-mediated resistance to other targeted therapies, such as the PI3K inhibitors (PI3Ki) alpelisib and pictilisib[15,49]. We generated both alpelisib and pictilisib-resistant subclones of T47D breast cancer cells, which displayed cross-resistance (Fig. 6a). Both drugs inhibited mTORC1 signaling in parental cells as evidenced by reduced phosphorylation of p70S6K[T389], 4E-BP1[T37/46], and ULK1[S757] (Fig. 6b). Consistent with previous reports[15], PI3Ki-resistant cells maintained mTORC1 signaling under PI3K inhibition as the underlying cause of resistance (Fig. 6b).

In line with suppression of autophagy by mTORC1 signaling, PI3Ki-resistant T47D clones failed to induce autophagic flux upon 2DG/DCA treatment (Fig. 6c). Of note, PI3K inhibition also stimulated LC3 processing and p62 degradation, indicative of autophagy induction—again only in parental, but not PI3Ki-resistant subclones (Fig. 6b). Autophagy induction by 2DG/DCA was restored upon mTOR inhibition, indicating a profound mTOR-dependent defect in the autophagy response of PI3Ki-resistant tumor cells. Consistent with our previous results, this translated into a massive mTOR-dependent induction of apoptosis and reduction of clonogenic growth by 2DG/DCA treatment (Fig. 6d, e), which extends our findings on mTOR-

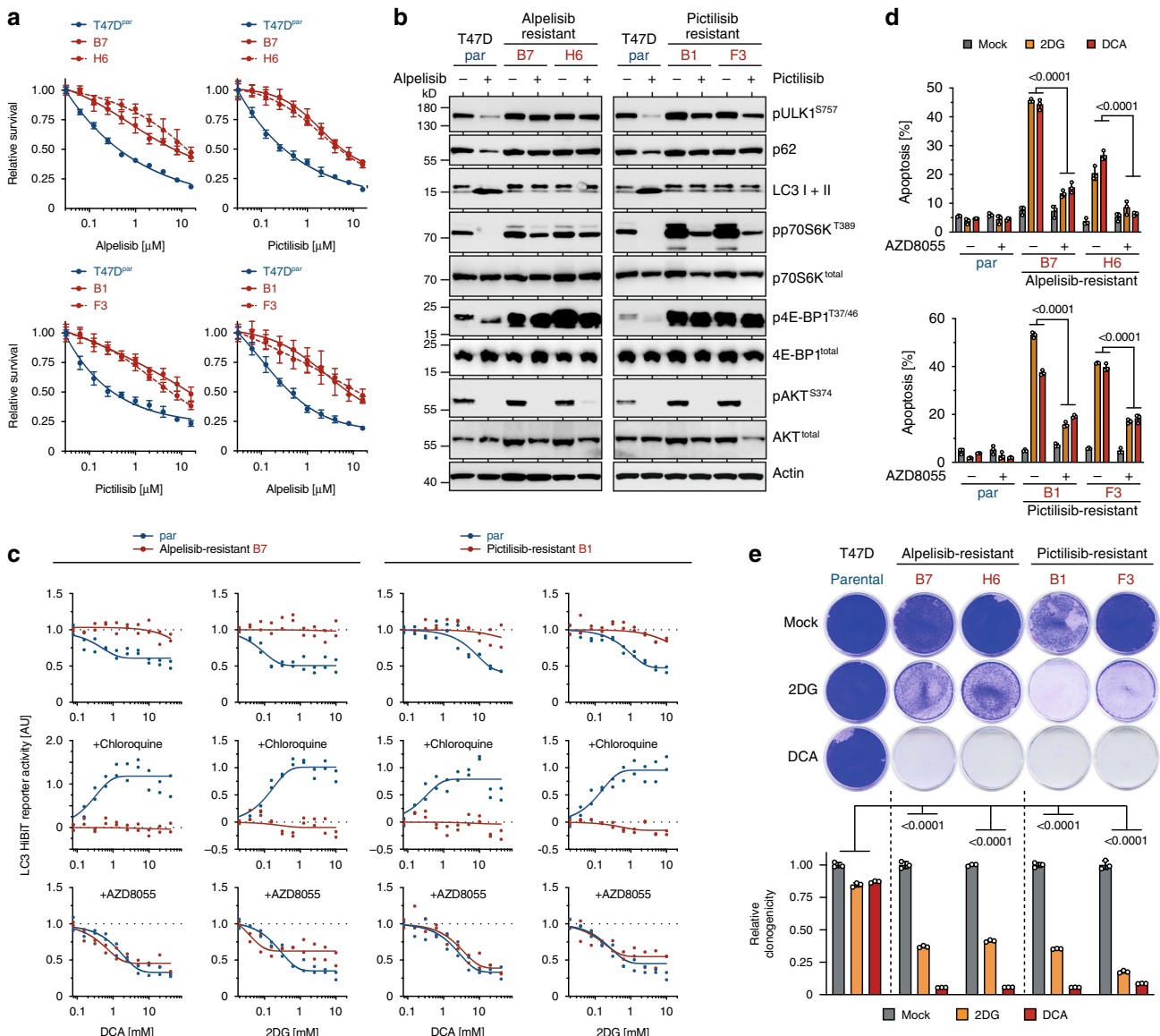

**Fig. 6 Metabolic vulnerability of tumor cells resistant to molecular targeted drugs.** T47D breast cancer cells were treated with increasing doses of alpelisib or pictilisib to obtain alpelisib-resistant (B7, H6) and pictilisib-resistant (B1, F3) subclones. **a** Cell viability of indicated T47D cells after 3 day treatment with alpelisib and pictilisib. Shown are mean ± SD, $n = 3$. **b** Western blot following treatment with alpelisib or pictilisib for 2 hours. **c** Autophagic flux assay. Parental and alpelisib/pictilisib-resistant LC3-HiBiT expressing T47D cells were pre-treated as indicated with chloroquine and AZD8055 for 48 hours and then with increasing doses 2DG/DCA for 6 hours. Shown is LC3-HiBiT reporter activity measured as luminescence normalized to untreated of $n = 2$ replicates. **d** Flow cytometry analysis for apoptosis (sub-G1). Shown are mean ± SD, $n = 3$, FDR $q$ values. **e** Clonogenic growth of parental and alpelisib/pictilisib-resistant T47D cells treated with 2DG/DCA. Shown are representative images and quantification as mean ± SD, $n = 3$, two-way ANOVA with Dunnett's multiple comparisons test.

induced metabolic vulnerability from classical chemotherapy to targeted therapy resistance.

**Metabolic inhibitors target CDDP-resistant cancer cells in vivo.** We proceeded to evaluate the efficacy of metabolic compounds for targeting tumor cells with mTOR-dependent drug resistance in vivo. To directly compare the effect of metabolic inhibitors on parental and drug-resistant tumor cells, we injected mice with a 1:1 mixture of H460[par] and H460[res] cells. To distinguish the two cell types in vivo, they were labeled with one of the two luciferases CLuc and GLuc, which have distinct substrate specificities yielding CLuc+H460[par] and GLuc+H460[res] cells. Both luciferases are actively secreted by the tumor cells, so that their concentration in

blood samples serves as an accurate measure of the number of viable tumor cells in the organism. Luciferase activity measurements of blood samples collected at regular time intervals during therapy provided a cell type specific longitudinal monitoring of tumor burden during therapy (Fig. 7a)[32]. Although growth of parental H460 cells was efficiently inhibited by CDDP treatment, growth of H460[res] cells was unaffected. Vice versa, DCA selectively inhibited the growth of H460[res] cells, resulting in a >40-fold depletion of H460[res] cells from the end-stage tumors. To confirm the observed changes in cellular abundance, tumor lysates from end-stage mice were analyzed for mTOR and autophagy markers (Fig. 7b). Relative to tumors from untreated mice, that are composed of both tumor cell types, tumors from the CDDP-treated

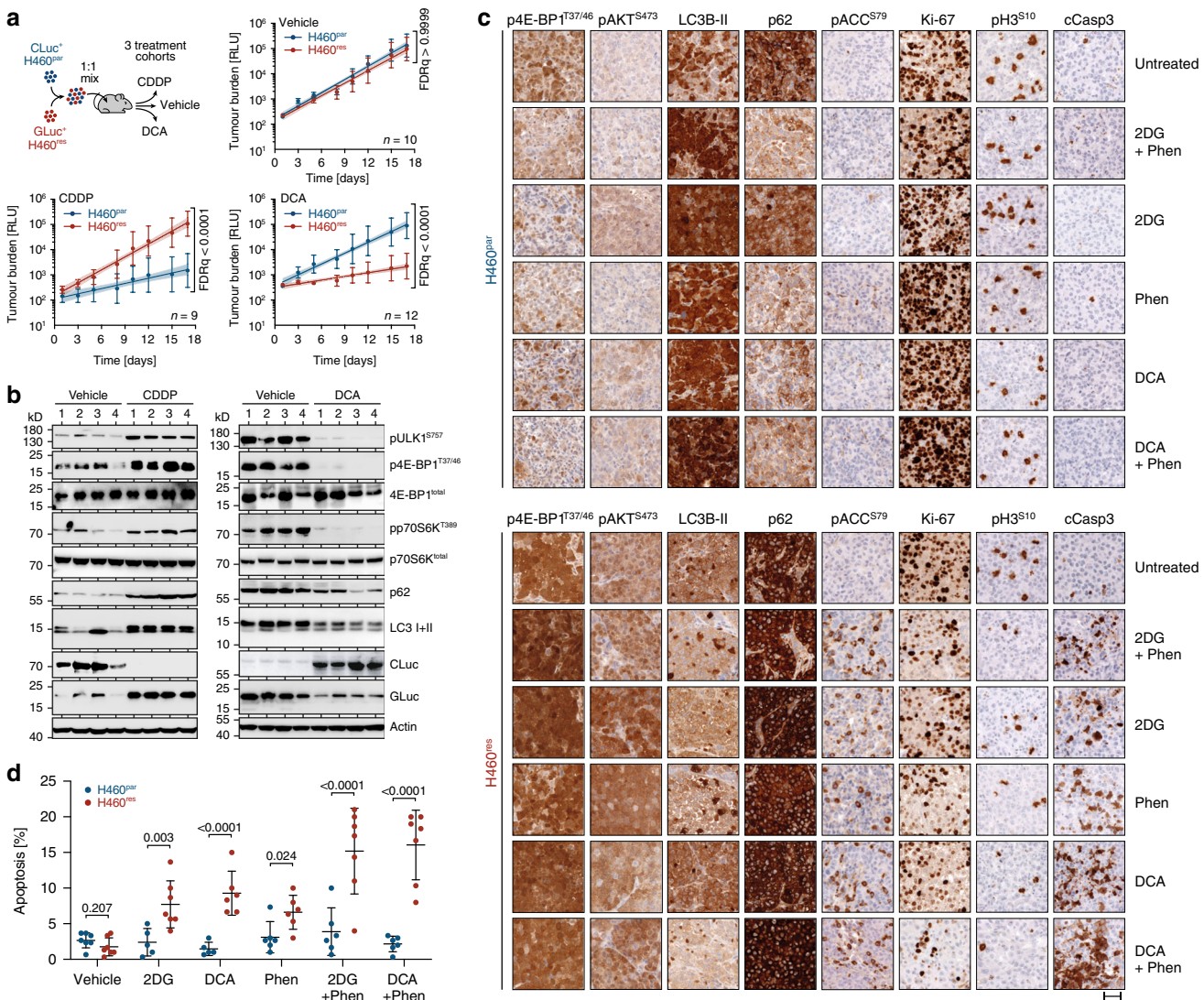

**Fig. 7 Metabolic inhibitors selectively target CDDP-resistant cancer cells in vivo. a, b** Mice were subcutaneously injected with a 1:1 ratio of H460[par] and H460[res] cells labeled with the secreted luciferases GLuc and CLuc, respectively, and treated as indicated. Tumor burden was quantified separately for each cell type by longitudinal luciferase activity measurement in blood samples. **a** Tumor growth curves show the tumor burden over time as mean ± SD and a linear regression curve with 95% CI, FDR q values. **b** Western blot of four tumors from each mouse cohort in **a**. **c**, **d** H460[par] and H460[res] cells were injected intravenously into immunodeficient mice. Developing tumors were treated with the indicated drugs for 3 days. Tissue sections were immunostained for the indicated proteins. Shown are representative images of tumors from each treatment cohort. Scale bar, 40 μm. **d** Quantification of apoptotic index (cleaved caspase-3) from n = 100 cells in three fields of view/mouse. Shown is the apoptotic index for individual mice, mean ± SD, FDR q values.

cohort displayed upregulation of mTOR signaling, defects in LC3 processing and p62 degradation (Fig. 7b, left panel), consistent with the CDDP-induced enrichment of H460[res] cells observed by luminescence-based tumor monitoring. In contrast, tumors from DCA-treated mice showed reduced mTOR signaling and reduced levels of LC3 and p62 (Fig. 7b, right panel), consistent with depletion of H460[res] cells by DCA. Thus, proliferation of CDDP-resistant H460[res] tumor cells was selectively inhibited by DCA therapy in vivo, resulting in the effective depletion of CDDP-resistant cells from heterogenous tumors.

To analyze the acute in vivo response to different metabolic inhibitors in more detail, we treated cohorts of mice bearing either H460[par] or H460[res] tumors for 3 days with 2DG, DCA, and Phen as single agents or drug combinations (Fig. 7c, Supplementary Fig. 6). Immunohistochemical analysis of tumor sections showed overall higher 4E-BP1[T37/46] and AKT[S473] phosphorylation in untreated H460[res] compared with H460[par] tumors, indicative of elevated basal mTOR signaling in CDDP-resistant tumors in vivo. Of note, similar as has been observed in vitro (Figs. 2a, 5b), mTOR signaling in H460[par] tumors remained low or decreased under treatment, whereas it was maintained or even increased in treated H460[res] tumors (Fig. 7c, Supplementary Fig. 6).

Drug treatment of H460[par] tumors resulted in increased LC3B-II staining, LC3 puncta formation and p62 degradation, as immunohistochemical markers for autophagy induction[39]. H460[res] tumors displayed elevated p62 levels that failed to decrease upon drug treatment and instead showed increased levels of the AMPK-triggered ACC[S79] phosphorylation as a marker for loss of energy charge. As a consequence, treated H460[res] tumors, unlike H460[par] tumors, showed a reduction in proliferation rate as evidenced by a reduced number of cells positive for Ki67 or phospho-H3[S10].

Most importantly, cleaved caspase-3 as a marker for apoptosis was significantly upregulated by single 2DG or DCA treatment in H460[res] tumors (Fig. 7c, d, Supplementary Fig. 6). Phen alone increased apoptosis only slightly, but strongly potentiated the apoptotic effect of 2DG/DCA, even though the 2DG/DCA dose in the combination was reduced to 50% (Fig. 7d). We conclude that therapy-resistant xenograft tumors with upregulated mTOR signaling fail to activate autophagy in response to pharmacologically induced energy stress and are therefore selectively targeted in vivo by multiple different metabolic drugs.

## Discussion

Changes in cellular energy metabolism are a hallmark of cancer, but as these alterations are mostly regulatory in nature and only rarely caused by mutations in metabolic enzymes, selective targeting of cancer cells with metabolic inhibitors is challenging and complicated by dose-limiting toxicity to normal tissues[27]. In our study, we have revealed that constitutive upregulation of mTOR signaling not only drives resistance of cancer cells to chemotherapy and targeted drugs, but also generates a druggable metabolic vulnerability that can be therapeutically exploited. Heterogenous xenograft tumors in mice composed of a mixture of therapy-naive and drug-resistant clones were selectively depleted of the resistant tumor cells using treatment protocols containing the anti-Warburg drugs 2DG or DCA and the anti-diabetic biguanide phenformin alone or in combination (Fig. 7). The drug-resistant and metabolically vulnerable cells displayed increased constitutive phosphorylation of many mTORC1 targets (Fig. 2a–c), inhibition of mTOR kinase, or specifically mTORC1, protected from metabolic inhibitors (Figs. 2d–g, 5b, 6d), and experimentally enforced activation of mTORC1 was sufficient to induce metabolic hypersensitivity (Figs. 2h–l, 5c, d). The intricate link between mTORC1 signaling and metabolic vulnerability strongly suggests that markers of mTORC1 signaling such as 4E-BP1 or p70S6K phosphorylation, or (rare) constitutively activating mutations in mTOR or pathway regulators[11,12,51,52], might serve as biomarkers to identify metabolically druggable tumors.

Given that the toxicity of metabolic drugs was linked to an mTORC1-induced autophagy defect (Figs. 3 and 5a), mediated at least in part by increased ULK1[S757] phosphorylation (Figs. 2 and 6), signs of an mTORC1-dependent autophagy deficiency such as pULK1[S757] might also signal metabolic susceptibility. Of note, compared with normal lung, ULK1[S757] phosphorylation is increased in NSCLC patients, predominantly lung adenocarcinoma (LUAD), and associated with shortened overall survival[53]. p62, which links LC3 and ubiquitinated substrates and ultimately becomes degraded in autolysosomes, can also serve as an index of autophagic degradation[39,54]. Different from therapy-naive tumors, we observed immunohistochemically increased p62 protein levels in CDDP-resistant tumor cells and these failed to decrease under all treatments with metabolic inhibitors (Fig. 7c), pinpointing p62 as a potential biomarker in immunohistochemistry. Furthermore, although autophagy genes are not among the most common genes hit by somatic point mutations, combined monoallelic deletions in multiple autophagy genes, including *MAP1LC3B* (LC3) and *BECN1*, have been identified in cancer types such as ovarian and breast cancer by haploinsufficiency network analysis, suggesting that autophagy status might also be inferred from genome sequencing data[55,56]. However, whether autophagy can serve as a suitable biomarker is under scrutiny as the autophagy status of tumors is still considered notoriously difficult to assess accurately in a clinical setting[39].

In our experiments, the mTOR-induced autophagy defect sensitized to metabolic inhibitors with very distinct modes of action. 2DG is a glucose analog that is taken up by cells through glucose transporters and phosphorylated by hexokinase. It cannot be further metabolized in the glycolysis pathway and therefore considered a glycolysis inhibitor. Newer studies on the basis of pulsed stable isotope-resolved metabolomics, however, suggest that 2DG does not directly block glycolysis but rather interferes with phosphate and ATP metabolism[57]. DCA, a structural analog of pyruvate, functions as an inhibitor of pyruvate dehydrogenase kinase (PDHK), thus increases pyruvate dehydrogenase complex activity and stimulates flux from glycolysis into the tricarboxylic acid cycle[58]. PDHK as the relevant target of DCA in our experiments was confirmed with AZD7545, a novel PDHK inhibitor that selectively killed CDDP-resistant H460 cells at low micromolar concentration (Supplementary Fig. 4)[45]. Although the impact of 2DG and DCA on cancer cell metabolism is still incompletely understood, both compounds are widely considered as anti-Warburg drugs that interfere with the central role of glycolysis in cancer cells[26]. In contrast, the anti-diabetic biguanides metformin and phenformin impair oxidative phosphorylation (OXPHOS) by acting as mitochondrial complex I inhibitors[47,59–61]. Despite some controversies regarding the precise mechanism, recent in vivo evidence supports that inhibition of mitochondrial complex contributes to the anti-tumorigenic effect of metformin, consistent with similar antitumor efficacy of other complex I inhibitors[47,62–67].

The hypersensitivity to a variety of metabolic inhibitors with diverse modes of action suggests a more general susceptibility of drug-resistant cancer cells to metabolic perturbation, rather than an acquired dependence on single metabolites or metabolic pathways. This is in line with autophagy being a catabolic process whereby intracellular components are captured, degraded, and recycled in lysosomes to provide all kinds of different metabolites required to temporarily sustain metabolism and maintain metabolic homeostasis during interruptions in extracellular nutrient availability[8]. For example, starved Ras-driven lung cancer cells heavily rely on autophagy to recycle intracellular nutrients into central carbon metabolism, maintain energy charge and nucleotide pools and keep reactive oxygen species under control[68]. Furthermore, autophagy induction is a side effect of many cancer therapies (chemotherapy, radiotherapy, and targeted agents) and thus promotes survival during the stress of therapies[8]. As such, pharmacological inhibition of autophagy has been proposed as a valid strategy to enhance the efficacy of therapies and to avoid treatment resistance[69–71]. Although autophagy induction has also been observed following treatment of cancer cells with metabolic inhibitors like metformin and 2DG, this has mostly been considered an antitumor effect contributing to tumor cell death[72,73]. In contrast, we demonstrate here that induction of autophagy is limiting the therapeutic effects of metabolic inhibitors. Furthermore, due to the ability to provide numerous different metabolites, it is well conceivable that autophagy can mitigate the antitumor effects of many different types of metabolic perturbation, so that the observed vulnerability of drug-resistant cancer cells with mTOR-induced autophagy defects might not be restricted to compounds targeting glycolysis, PDHK or OXPHOS.

Notably, we observed mTOR-dependent metabolic vulnerability in tumor cells from different NSCLC subtypes, Kras- and EGFR-driven LUAD. Confirming our observations from cell culture and mouse models, mTOR upregulation has previously been observed in Kras-mutant lung cancer patients that have relapsed following platinum-based chemotherapy[35]. Although aberrant MAPK signaling is a general feature of LUAD, mutations in signal transducers like Kras are commonly associated with smoking, whereas mutations in receptor tyrosine kinases such as EGFR are more frequent in non-smokers[74]. Development of mTOR-dependent vulnerability in both Kras- and EGFR-mutant LUAD cells suggests that this phenotype could be a

prevailing characteristic of chemotherapy-relapsed LUAD or MAPK-driven cancers in general, irrespective of the underlying pathway mutation. Of note, mTOR upregulation is also considered a cause of resistance to EGFR-inhibitors in lung cancer patients[13,75], implying that they might also display metabolic vulnerability.

In light of mTOR as a widespread driver of therapy resistance, there is considerable hope for targeting cancer drug resistance with mTOR inhibitors[4]. However, at present there is little evidence that upregulation of mTOR in a resistance setting causes an mTOR-dependence and sensitizes resistant tumor cells to monotherapy with mTOR inhibitors. In fact, in our study H460[res] cells were equally insensitive to mTORC1-selective or dual mTOR kinase inhibitors as parental H460 cells (Supplementary Fig. 1). Instead, many clinical trials are underway combining either chemotherapies or specific targeted therapies with mTOR inhibitors to overcome or prevent the emergence of mTOR-mediated resistance[4,71]. Likewise, studies have proposed mTOR inhibition as a therapeutic strategy to target cancer metabolism and even suggested combining mTOR inhibitors with metabolic compounds to target cancer cells more effectively[76]. In contrast, our study highlights mTOR as a double-edged sword, which on one hand induces resistance to chemotherapy and targeted drugs, but on the other drives metabolic vulnerability. Combining mTOR inhibitors with metabolic compounds would thus be counterproductive. In fact, our experiments show that metabolic compounds are most effective in cancer cells in which autophagy is blocked by high-level mTOR signaling and that mTOR inhibition reduces their anti-cancer activity (Figs. 2 and 5).

In summary, our study reveals an exciting, ambivalent function of mTOR, which renders tumor cells resistant to chemotherapy and targeted cancer drugs, but simultaneously suppresses autophagy, an essential survival mechanism for coping with therapy-induced metabolic perturbations. Intriguingly, our results imply that metabolic inhibitors might be most effective on those tumor cells that are otherwise therapy-resistant and that signs of mTOR signaling such as ULK1[S757] phosphorylation are potential biomarkers for metabolically vulnerable tumors.

## Methods

**Cell culture**. Cell lines were obtained from the American Tissue Collection Center (ATCC) and grown in high-glucose DMEM (HCT116, U2OS Hela, T47D) or RPMI1640 medium (H460, H1975) supplemented with 10% fetal bovine serum, 100 U ml[−1] penicillin and 100 μg ml[−1] streptomycin at 37 °C with 5% CO$_2$. All cell lines were authenticated by short tandem repeats profiling and tested negative for mycoplasma contamination. Murine $Kras^{G12D/wt}$;$Trp53^{\Delta/\Delta}$ lung adenocarcinoma cells from Adeno-Cre infected $Kras^{LSLG12D/wt}$;$Trp53^{flox/flox}$ mice[33] were cultured in RPMI1640 medium supplemented as above. CDDP[naïve] mouse tumor cells were derived from untreated tumors, CDDP[res] cells from tumors that had relapsed after an initial CDDP treatment. Chemotherapy drugs and inhibitors were obtained from Sigma-Aldrich unless indicated otherwise and used at the following concentrations: CDDP 1 μg ml[−1], oxaliplatin 5 μg ml[−1], carboplatin 5 μg ml[−1], etoposide 1 μM, doxorubicin 0.04 μg ml[−1], RITA (Merck) 1 μM, DCA 20–40 μM, 2DG 10–30 mM, metformin 1.25–4 mM, phenformin 60–125 μM, AZD8055 (Selleckchem) 0.2–1 μM, Rapamycin (Selleckchem) 250 nM, Everolimus (Absource Diagnostic) 250 nM, AZD7545 (Selleckchem) 10 μM, chloroquine 25–100 μM, 3-methyladenine (Calbiochem) 5 mM, and SBI-0206965 (Biovision) 5 μM. For starvation experiments, cells were grown in Hank's Balanced Salt Solution (Sigma) for up to 5 days.

**Generation of drug-resistant cell lines**. CDDP-resistant H460[res] cells were generated by CDDP dose escalation (5 nM–2.56 μM). CDDP-resistant H1975 cells were generated by 4–5 rounds of CDDP treatment with 0.5 μg ml[−1] for 3 hours and five rounds with 1 μg ml[−1] for 3 days. RITA-resistant cells were generated by continuous treatment with 1 μM RITA or dose escalation from 5 nM to 2.5 μM. PI3Ki-resistant T47D cells were generated by dose escalation from 20 nM to 2.5 μM alpelisib or pictilisib, respectively. Resistant cell lines were treated continuously with 1 μg ml[−1] CDDP, 1 μM RITA, 1 μM alpelisib or 1 μM pictilisib. CDDP[res] mouse tumor cells were maintained in the presence of 0.5 μg ml[−1] CDDP. Resistant cells were cultured in the absence of drug for at least a week before an experiment.

**Cell viability assays**. Cell viability in response to treatment was measured using the CellTiter-Glo assay (Promega) according to manufacturer's instructions. In brief, cells were seeded in white-walled 96-well plates overnight and 2−3 replicate wells were treated with inhibitors diluted in 80 μl medium the next day. After 72 h of treatment, 80 μl CellTiter-Glo reagent was added to wells und luminescence was recorded using an Orion II luminometer (Berthold). Background signal from empty wells was subtracted and luminescence was normalized to untreated control wells. Dose–response curves were fitted to the data points with GraphPad Prism software (inhibitor vs. response–variable slope four parameter model) and used to determine the 50% inhibitory concentration (IC50) with 95% confidence intervals.

**Clonogenic growth assays**. Cells were plated overnight, treated and cultivated for ~10 days. mTOR inhibitors were applied 3 days before treatment with other drugs and discontinued after a few days of co-treatment with the first medium change, as continuous treatment with mTOR inhibitors alone showed a cytostatic effect (Supplementary Fig. 1). Plates were fixed in 70% ethanol overnight and stained for 30 minutes with crystal violet solution (Sigma-Aldrich HT90132 diluted 1:20 in 20% ethanol), washed in tap water and air-dried. For quantification, acetic acid (20 vol%) was added and incubated for 10 min on a shaker. The optical density of a 1:20 dilution in water was measured in triplicates at 590 nm using a Cytation 3 plate reader (Biotek). Following background subtraction, the OD590 was normalized to untreated reference samples yielding relative clonogenicity values.

**Live-cell imaging**. Real-time monitoring of tumor cell proliferation was performed using an IncuCyte S3 Live-Cell Analysis System (Sartorius). Cells were seeded on 96-well plates overnight and treated with drugs/inhibitors on the next day. Every 2 hours we recorded four phase contrast images per well at ×10 magnification, with three replicate wells per treatment condition. Confluence analysis was performed with IncuCyte S3 2018A software in Phase Object Confluence mode, using a segmentation score of 0.7 and excluding objects smaller than 500 μm$^2$.

**Apoptosis analysis**. Cell culture supernatant was collected. Adherent cells were trypsinised, combined with cell culture supernatant, and washed in 5 ml phosphate-buffered saline (PBS). Cell pellet was resuspended in 1 ml PBS and fixed by drop-wise addition of 10 ml ice-cold 90% ethanol while vortexing. After fixation overnight, cells were stained with 10 μg ml[−1] propidium iodide supplemented with 100 μg ml[−1] RNase A. Cells were analyzed for sub-G1 content on an Accuri C6 Plus cytometer (BD Bioscience).

**Autophagy assays**. For analysis of autophagosome formation, cells were transduced with a retroviral DsRed-LC3-GFP expression construct, which has GFP separated from the C-terminus of DsRed-LC3 by a recognition site for the autophagic protease ATG4 (pQCXI Puro DsRed-LC3-GFP, Addgene plasmid # 31182)[41]. Under basal conditions, DsRed-LC3-GFP-expressing cells show diffuse cytoplasmic green fluorescence that, upon autophagy induction, is lost and shifted to red-fluorescent puncta marking autophagosomes[41]. DsRed-LC3-GFP-expressing cells were seeded on an eight-well chamber slide (Sarstedt) and treated with 2DG (20 mM) or DCA (30 mM) for 24 hours. Cells were fixed with 3.7% paraformaldehyde (Sigma-Aldrich) in PBS, pH 7.4, and stained with 200 nM DAPI (Molecular Probes) for 45 min at RT. Images were obtained with the Leica DM4 B Fluorescence microscope at ×63 magnification with oil immersion using three distinct fluorescence channels (DAPI, GFP, DsRed) with equal exposure time and gain for all samples. Merged images of all three channels were generated and analyzed with the Aperio Software (Leica). For quantification, red-fluorescent LC3 puncta were counted in 100 cells per well.

For autophagy flux analysis, we used the autophagy LC3-HiBiT Reporter vector (Promega, GA2550) encoding a fusion protein consisting of human LC3B, a small N-terminal 11 amino acid HiBiT tag, and an intervening proprietary spacer region that enhances reporter specificity for the autophagic pathway. It utilizes the constitutive HSV-TK promoter with PyF101 enhancer for low-to-moderate expression of the reporter in mammalian cells. Cells were transfected with the reporter vector using Lipofectamine 2000 (Invitrogen) and selected with Geneticin (Gibco) at 800 μg ml[−1]. Stably transfected LC3-HiBiT reporter cell lines were maintained in 400 μg ml[−1] Geneticin and seeded on 96-well plates for analysis. Increasing concentrations of metabolic drugs (2DG, DCA, Met, Phen) were applied the next day and LC3-HiBiT reporter activity was measured after 6 hours using the Nano-Glo HiBiT Lytic Detection System (Promega, N3040) and an Orion II luminometer (Berthold) according to the manufacturer's protocol. A decrease in LC3-HiBiT reporter activity reflects autophagic LC3 degradation and serves as a measure of autophagy induction. In other experiments, chloroquine was used to block autophagosome-lysosome fusion and prevent LC3-HiBiT reporter degradation, which resulted in reporter accumulation. Basal autophagy levels were assessed by LC3-HiBiT reporter accumulation in response to increasing doses of chloroquine for 6 hours. A metabolic drug-induced increase of reporter activity in the presence of chloroquine (50 μM) therefore served as a measure for autophagic flux stimulation. To assess the impact of endogenous mTOR activity on drug-induced autophagy, LC3-HiBiT reporter assays were conducted in cells treated with AZD8055 (0.5 μM). Chloroquine and AZD8055 treatment was started 48 hours

before the 6 hour-treatment with metabolic drugs. Following background subtraction, luminescence signals were normalized to untreated samples.

**RNAi**. Cess were plated at 50–80% confluence 1 day before transfection. siRNAs were obtained from Dharmacon and used for transfection with Lipofectamine RNAiMAX reagent (Invitrogen) at a final concentration of 20 nM according to the manufacturer's instructions. siRNA sequences: TSC1/2-si1: CGACACGGCUGA UAACUGA and GCAUUAAUCUCUUACCAUA, TSC1/2-si2: CGGC UGAUGUUGUUAAAUA and GGAUUACCCUUCCAACGAA, mTOR-si1: GGCCAUAGCUAGCCUCAUA, mTOR-si2: GCAGAAUUGUCAAGGGA UA, ATG7-si1: GAUCUAAAUCUCAAACUGA, ATG7-si2: CCAACACACU CGAGUCUUU, ATG7-si3: GCCAGAGGAUUCAACAUGA, ATG14-si1: CGG GAGAGGUUUAUCGACA, ATG14-si2: GCUCAGAAAUGCGACGG, ATG 14-si3: CAUUAUGAGCGUCUGGCAA, ATG14-si4: GAUAAAUCAUUCAGA GGUU, FIP200-si1: GGAGUGGGCUGGUGCUUUA, FIP200-si2: AAACUA CGAUUGACACUAA, FIP200-si3: GCAAAGAAAUUAGGGAAUC, FIP200-si4: UAAACUUGACGGACUAAUA, RUBCN-si1: CGGGAUUAGAUUAGCGUUA, RUBCN-si2: CAUGAGACCUCGAGCGAAC, RUBCN-si3: GAGAAUAGGAC UACGUCAU, RUBCN-si4: CCCUGGAAGCUGAGCGGAA, RPTOR-si1: UGGC UAGUCUGUUUCGAAA, RPTOR-si2: UGGAGAAGCGUGUCAGAUA, RICTOR-si1: GAAGAUUUAUUGAGUCCUA, RICTOR-si2: GGGAAUACAACUCCAAA UA, non-silencing si (nsi): UGGUUUACAUGUUUUCUGA.

**Lentiviruses**. Lentiviral vectors for FLAG-Rheb1 (Flag pLJM1 Rheb1; Addgene plasmid #19312), Gaussia luciferase (GLuc) and Cypridina Luciferase (CLuc)[32,77] were transfected into 293 T cells together with packaging plasmids (pMD2.G Addgene plasmid #12259 and psPAX2 Addgene plasmid #12260) using the calcium-phosphate method. Supernatants were collected on the second and third day after transfection and supplemented with polybrene (8 μg ml$^{-1}$) for infection of target cells. Transduced cells were selected with puromycin (2 μg ml$^{-1}$) for 5 days and maintained in 1 μg ml$^{-1}$ puromycin.

**PiggyBac transposons**. Plasmids encoding mTOR mutants L1460P, I2500F, and S2215Y (Addgene #69006, 69014, 69013) were subcloned via NotI into the PiggyBac transposon expression vector PB-EF1α-MCS-IRES-Neo (System Biosciences). The transposon vector was co-transfected with pCMV-HAhyPBase[78] into H460 cells using Lipofectamine 2000 (Thermo Fisher) and Neomycin-resistant cell clones were selected and expanded in the presence of Geneticin (400 μg ml$^{-1}$).

**CRISPR-Cas9**. For generating ATG7 knockout cell lines, H460 cells were infected with plentiCRISPRv2 (Addgene #52961) lentiviral particles encoding the two ATG7-directed sgRNAs ATG7sg1 AGAAATAATGGCGGCAGCTA and ATG7sg2 TGCCCCTTTTAGTAGTGCCT. After puromycin selection (1 μg ml$^{-1}$), genomic DNA from single cell clones were analyzed by PCR and Sanger sequencing for the presence of CRISPR-induced indel mutations in ATG7 using the following primers GAATGGTGGGTGCAAGGTCC and GACAAGGCTGTCTCTTTCTAA AGG. For generating ATG14, FIP200, and RUBCN knockout cells, H460par cells were transfected with pSpCas9(BB)-2A-Puro (pX459) V2.0 (Addgene #62988) plasmids containing the sgRNA sequences: ATG14sg1 TGAAGGCCTTCTCAAAAC CA, ATG14sg2 AGCTTTACAGTCGAGCACAA, ATG14sg3 AGAAAAAGGAGA AGATTCAG, ATG14sg4 CTCGATTGGAAAAATGACAG, ATG14sg5 CCAATCG AGGAAGTAAAGAC, FIP200sg1 CTGGTTAGGCACTCCAACAG, FIP200sg2 AG GAGAGAGCACCAGTTCAG, FIP200sg3 AACCTCATTTCCCAAGTCAG FIP200-sg4 GATACCGCAGATGCTGAAAG, FIP200sg5 TCAAGATAGACCTAATGATG. Transfected cells were selected with puromycin (1 μg ml$^{-1}$) and validated for successful knockout by western blot.

**Western blotting**. Cells were lysed in NP-40 Lysis Buffer (50 mM Tris-HCl, 150 mM NaCl, 5 mM ethylenediaminetetraacetic acid (EDTA), 2% NP-40, pH 8.0) supplemented with protease (complete ULTRA tablets EASYpack, Roche) and phosphatase inhibitor (PhosSTOP, Roche). Protein yield was determined by Bradford assay (Bio-Rad). Total protein (5–50 μg) was separated on NuPAGE SDS Gels (Life Technologies) and tank-blotted to PVDF membranes. Following blocking in tris-buffered saline with polysorbate 20 (TBST; 5 mM Tris, 15 mM NaCl, 0.1% Tween 20, pH 7.5) with 5% nonfat dry milk, membranes were incubated with primary antibodies diluted in TBST/5% nonfat dry milk and incubated overnight at 4 °C. Antibodies: Gaussia Luciferase GLuc (1:1000, #401 P, Nanolight), Cypridina Luciferase CLuc (AA1-168) (1:200, ABIN1605705, antikoerper-online.de), cleaved PARP (Asp214) (1:1000, #9541, Cell Signaling), FANCD2 (1:500, sc-20022, Santa Cruz), FLAG-tag (1:1000, F1804, Sigma-Aldrich), phospho-ULK1 (Ser757) (1:1000, #14202, Cell Signaling), ULK1 (1:1000, R600 #4773, Cell Signaling), p62/SQSTM1 (1:1000, P0067, Sigma), LC3B (LC3-I/II) (1:1000, ab48394, Abcam), mTOR (1:1000, 7C10 #2983, Cell Signaling), phospho-p70S6Kinase (Thr389) (1:1000, 108D2 #9234, Cell Signaling), p70S6Kinase (1:500, H9 sc-8418, Santa Cruz), phospho-4E-BP1 (Thr37/46) (1:1000, 236B4 #2855, Cell Signaling), 4E-BP1 (1:200, R-113, sc-6936, Santa Cruz), phospho-S6 Ribosomal protein (Ser240/244) (1:1000, #2215, Cell Signaling), S6 Ribosomal protein (1:1000, 5G10 #2217, Cell Signaling), phospho-Akt (Ser473) (1:1000, D9E #4060, Cell Signaling), Akt (1:1000, #9272, Cell Signaling), Hamartin/TSC1 (1:1000, D43E2 #6935, Cell Signaling), Tuberin/

TSC2 (1:1000, D93F12 #4308, Cell Signaling), Raptor (1:1000, 24C12 #2280, Cell Signaling), Rictor (1:1000, 53A2 #2114, Cell Signaling), phospho-AMPKα (Thr172) (40H9) (1:1000, #2535, Cell Signaling), AMPKα (1:1000, 23A3 #2603, Cell Signaling), phospho-Acetyl-CoA carboxylase (Ser79) (1:1000, #3661, Cell Signaling), Acetyl-CoA carboxylase (1:1000, C83B10 #3676, Cell Signaling), phospho-pyruvate dehydrogenase E1-alpha subunit (S293) (1:500, ab92696, Abcam), pyruvate dehydrogenase E1-alpha subunit (1:500, ab110330, Abcam), ATG7 (D12B11) (1:1000, #8558, Cell Signaling), ATG14 (1:1000, #5504, Cell Signaling), FIP200 (1:1000, D10D11 #12436, Cell Signaling), RUBCN (1:1000, D9F7 #8465, Cell Signaling), β-Actin (AC-15) (1:10.000, ab6276, Abcam). Proteins were detected with a secondary antibody: sheep anti-mouse IgG-HRP (1:5000, #NA9310, GE Healthcare), goat anti-mouse IgG-HRP (1;5000, #A16084, Thermo Fisher Scientific), donkey anti-rabbit IgG-HRP (1:5000, #NA9340, GE Healthcare) and WesternBright ECL Substrat Sirius kit (Biozym).

**Animals**. All xenograft experiments were performed according to the German Animal Welfare Act (Deutsches Tierschutzgesetz) and were approved by the Regional Board Giessen. All mice were bred and maintained under specified pathogen-free conditions at a room temperature of 22 ± 1°, a relative humidity of 50 ± 10% and a 12/12 dark/light cycle. Mice were kept in groups of 3–7 and provided continuously with sterile water and chow pellets. Animal group sizes were calculated by an a priori power analysis. All experiments were done in 6–12 week-old, male and female C;129S4-Rag2$^{tm1.1Flv}$;Il2rg$^{tm1.1Flv}$/JThst mice, kept under specified pathogen-free conditions.

For monitoring the in vivo growth of H460$^{par}$ and H460$^{res}$ cells in a competitive setting, H460$^{par}$ cells were labeled with CLuc lentivirus and H460$^{res}$ cells with GLuc lentivirus[32]. CLuc and GLuc luciferases are secreted by tumor cells and their activity in blood samples quantitatively reflects the total amount of viable tumor cells in the mouse[32,79]. Both cell types were mixed in a 1:1 ratio and injected subcutaneously into mice (n = 36). Mice were randomly allocated to three treatment cohorts of n = 12 mice each. Luciferase expression was induced with doxycycline, continuously supplied in drinking water in darkened bottles at a concentration of 1 mg ml$^{-1}$ in water with 2% sucrose. Drinking water was changed every third day. CDDP (7 mg kg$^{-1}$ body weight) was administered twice intraperitoneally in 0.9% NaCl with 7 days pause. DCA (500 mg kg$^{-1}$ body weight) was administered orally once a day. Control mice received 0.9% NaCl as vehicle control. For monitoring of tumor growth, 10 μl of blood was obtained by tail vein puncture and mixed directly with 2 μl of 0.125 IU ml$^{-1}$ heparin. Plasma was collected by centrifugation (15 min, 3600 × g, 4 °C). All collected plasma samples were stored at −20 °C and measured together at the end of the experiment with a single batch of reagents. For luciferase activity measurements in the Orion II luminometer (Berthold), plasma was diluted 1:10–1:1000 with PBS. Each diluted sample (5 μl) was measured by injection of 100 μl coelenterazine (stock from PJK, Germany, diluted 1:200 dilution in PBS) or 25 μl vargulin reagent (stock from NEB diluted 1:200 in Biolux Cypridina Luciferase Assay Buffer prediluted 1:5 in PBS). Animals that did not develop tumors (two mice from control cohort) or showed an elevated blood luminescence already at the first day of measurement (three mice from CDDP cohort) were excluded from the analysis.

For analyzing short-term responses to multiple metabolic drugs in parallel, mice were intravenously injected with either H460$^{par}$ (n = 39 mice) or H460$^{res}$ cells (n = 50 mice). After 2 weeks of tumor growth, animals were treated for 3 days. DCA and phenformin diluted in sterile water (B. Braun) were administered orally once daily at 500 mg kg$^{-1}$ or 150 mg kg$^{-1}$ body weight, respectively. 2DG was administered intraperitoneally at 1000 mg kg$^{-1}$ also diluted in sterile water (B. Braun). In combination with phenformin (150 mg kg$^{-1}$), DCA and 2DG were administered at 250 mg kg$^{-1}$ or 500 mg kg$^{-1}$ body weight, respectively. Three days after start of treatment, animals were killed and their lungs were histologically analyzed. Animals that reached humane endpoints before the planned end of the experiment (two in the H460$^{par}$, four in the H460$^{res}$ cohort) or failed to develop tumors (two in the H460$^{par}$, six in the H460$^{res}$ cohort) were excluded from the analysis.

**Immunohistochemistry**. For immunohistochemistry, we performed heat-induced epitope retrieval with citrate (for p62 and LC3B-II), EDTA (for cleaved Caspase-3 and phospho-ACC) or Trilogy (for phospho-Histone H3, Ki67 and phospho-AKT). Staining was performed on a Dako Autostainer Plus. Sections were incubated for 45 min with the following primary antibodies: rabbit anti-Cleaved Caspase-3 Asp175 (1:200; Cell Signaling), rabbit anti-Phospho-Acetyl-CoA-Carboxylase Ser79 (1:500; Cell Signaling), rabbit anti-Phospho-Histone H3 Ser10 (1:200; Cell Signaling), rabbit anti-Ki67 (1:75; abcam ab15580), rabbit phospho-4E-BP1 Thr37/46 (1:1000, 236B4 #2855, Cell Signaling), rabbit anti-Phospho-AKT Ser473(1:25; Cell Signaling), mouse anti-p62 (1:20.000; abcam ab56416), rabbit anti-LC3B-II (1:2000; abcam ab48394). Sections were washed and incubated with Dako REAL EnVision Detection System according to the manufacturer's protocol and counterstained with hematoxylin. Slides were digitally imaged using the Leica Biosystems Aperio Versa 8 slide scanner. Immunostaining was quantified using Aperio image scope v12.3.2.8013 positive Pixel Count v9 algorithm. The positivity index was calculated as the ratio of positive pixels over the total number of pixels in regions used for quantification. The algorithm input settings were adjusted for each marker to reduce background staining.

**Statistics and data reproducibility**. For in vitro experiments, no statistical methods were used to predetermine sample size. For animal experiments, an a priori power analysis was performed to calculate the group size needed to measure an estimated effect size (Cohen's $d$) of 1.0 with sufficient statistical power ($\alpha = 0.05$, $1 - \beta = 0.80$). The experiments were not randomized and investigators were not blinded to allocation during experiments and outcome assessment. GraphPad Prism 8 and Microsoft Excel Software were used to generate all plots and perform statistical analysis. Experiments investigating the interaction of two variables (e.g., cell type and treatment) were analyzed using two-way analyses of variance followed by multiple comparison testing according to Dunnett, Tukey, or Sidak as indicated. Time course experiments were analyzed using multiple two-sided $t$ tests in combination with the false discovery rate approach (two-stage linear step-up procedure of Benjamini, Krieger, and Yekutieli). FDR $q$ values <0.05 are considered significant and are reported for the last time point or highest dose. When results from representative experiments are shown (micrographs, western blots, and clonogenic growth assays), these were replicated in at least two independent experiments with similar results. Only results depicted in Supplementary Fig. 3a, b were performed as a single experiment, as the conclusions were validated by the RNAi experiments depicted in (Supplementary Fig. 3c, d, f, g). In addition, all experiments with H460$^{res}$ cells were reproduced with an independently generated CDDP-resistant H460 cell clone and yielded similar results.

**Reporting summary**. Further information on research design is available in the Nature Research Reporting Summary linked to this article.

## Data availability

All data generated or analyzed during this study are included in this published article (and its supplementary information files). The reporting summary and editorial checklist for this article are available as a Supplementary file. The following source data are provided with this paper: uncropped images for Figs. 1b, 2a, b, c, f, h, j, 4b, 5b, d, 6b, 7b, Supplementary Figs. 1a, e, 3a, c, d, e, 4b, and 5c, numerical source data for Figs. 1c, e, g, h, 2e, g, k, 3a, b, c, e, 4f, g, 5a, c, 6a, c, d, e, 7a, d, Supplementary Figs. 1b, c, 4c, 5a, and 6. Source data are provided with this paper.

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

## Acknowledgements

We acknowledge Sigrid Bischofsberger, Antje Grzeschiczek, and Angela Mühling for experimental contributions and thank members of the laboratory for helpful discussion and advice. This work was supported by grants from the Deutsche Forschungsgemeinschaft (DFG WA 2725/2-1; TRR81/3 109546710 TPA10), German Center for Lung Research (DZL), Deutsche Krebshilfe (70112623), Universitätsklinikum Giessen & Marburg (UKGM 18/2019 MR).

## Author contributions

Conceptualization: N.G., O.T., M.W., and T.S.; methodology: N.G., S.E., C.B., O.T., M.W., and T.S.; investigation: N.G. with contributions from P.P., A.D., J.S., S.E., C.B.; formal analysis: N.G., P.P., A.D., S.E., M.W., and T.S.; resources: U.K., A.S., H.C.R.; writing—original draft: N.G., M.W. and T.S.; writing—review and editing: all.

## Funding

## Competing interests

The authors declare no competing interests.
