## [Peer Review File · Nature Communications]

Reviewers' Comments:

Reviewer #1:

Remarks to the Author:

The paper by Gremke et.al. reports that cancer cells with acquired drug resistance have enhanced mTOR signaling and that this leads to enhanced sensitivity to metabolic inhibitors due to the mTOR inhibiting autophagy. The authors use a variety of drug resistance models including models where resistance was acquired in vivo and use pharmacological and genetic methods to show that the effects are mediated through mTOR and through autophagy.

For the most part I found the data presented here to be convincing and I think that the authors conclusions are sound. My concerns are as follows.

1. Although I think the conclusions are correct, I am not so sure that this is a significant conceptual advance. As the authors note, mTOR is known to be a key driver of cancer drug resistance so it's not surprising that their resistant cells have upregulated mTOR signaling. mTOR is also well known to be the principle negative regulator of autophagy and autophagy is known to be a key determinant of cancer cell survival and autophagy inhibition is well known as a way to sensitize to anti-cancer treatments and metabolic dysfunction including block of glycolysis and with agents like metformin. So I think all these conclusions are what would be predicted. This is my main issue about the paper. If the authors can explain why I'm wrong, and what there is that is really new here, I'd perhaps reconsider.
2. The conclusion that these effects are due to autophagy is primarily based on inactivation of ATG7. However ATG7 is also involved in other related processes like LAP or LANDO (and completely independent processes such as p53 regulation). Confirmation that similar results are obtained by targeting other genes that are more specific for autophagy such as FIP200 or ATG14 would strengthen the conclusion that this is really due to autophagy.

Reviewer #2:

Remarks to the Author:

In the present MS, Gremke et al. identify a metabolic liability of cancer cells with acquired resistance to DNA damaging agents. They show that cultured human and mouse cancer cell lines evolve resistance to DNA damaging agents presumably by the activation of mTORC1 and elevating FancD2 levels. Interestingly, this mechanism of resistance sensitizes cancer cells to inhibitors of glycolysis. They further show that autophagy, a process blocked by mTORC1, is the prosurvival cellular function essential to sustain cancer cell viability in the presence of these inhibitors. The conclusions are sustained with solid data, based on several cancer cell lines and complemented with some experiments in mice, which are consistent with the results obtained in vitro. Certain aspects of the MS have been shown previously, albeit altogether, the message of the MS has provocative implications that would fit the expectations of readers of Nature Communications. Nevertheless, the MS may benefit from additional experimental support to some of the conclusions and implications.

-A premise of the MS is that 1- High PI3K/mTORC1 activity is a common mechanism of resistance to DNA damaging agents; and 2- Elevated FancD2 levels is a well-established mechanism to limit the efficacy of the agents. While there is published work to support for the first premise of the paper, acquired resistance to chemotherapeutic agents cannot be presented in such a simplistic manner.

-The striking essentiality of autophagy for cancer cell viability in the context of glycolytic inhibitors is well established. So is the prosurvival effect of autophagy upon exposure to DNA damaging agents, a concept that seems to contradict the data presented here. In addition, increased mTORC1 activity causing the inability to survive under energetic stress has been reported many

times. Several explanations have been proposed, including increased metabolic rate and energetic demand of the cell, limited nucleotide availability (which would be critical during DNA repair), and the block of autophagy. Hence, a critical experiment to support the whole model of the paper would be to ectopically express FancD2 in parental (DNA damage sensitive cells). According to the model of the MS, this should confer resistance to DNA damaging agents, and would make inhibition of PI3K-mTORC1 either with direct inhibitors (AZD8055, MLN0128, Torin1) totally innocuous during CDDP treatment. Although FancD2 is not the focus of the MS, its upregulation is an essential component of the model as it is presented.

-Besides the FancD2 connection, the lack of effect of inhibition of mTORC1 in cancer cell lines (Figure 2d and other panels) is rather surprising, more so considering the strong upregulation of the pathway in resistant cells. Acute response to mTORC1 inhibition (cell cycle arrest) should complement the end-point readouts presented. Additionally, readouts of DNA damage, such as gamma-H2AX in parental and resistant cells with and without pharmacological inhibition of mTORC1 would complement the data and support the model.

-Is inhibition of autophagy a sensitizer also for DNA damaging agents in the systems utilized in the MS?

-The in vivo experimental system used in Figure 1g is not exploited in Figure 7. Can the authors explain why?

-Pharmacological inhibition of mTOR and genetic activation of mTOR complex 1 are presented. To support the notion that mTORC1 (and not mTORC2) is responsible for the resistance to DNA damage and for the sensitivity to 2DG and DCA, the effect of rapalogs should also be tested in some of the experiments presented, such as the assay in Figure 2d.

Reviewer #3:

Remarks to the Author:

The manuscript by Gremke et al. provides a mechanism in which mTOR simultaneously promotes cancer cell resistance to chemotherapy and inhibits autophagy which is a necessary process for tumour cells to cope with metabolic perturbations induced by chemotherapies. This work demonstrates that cisplatin-resistant cells exhibit metabolic vulnerabilities that are mediated by mTORC1. Indeed, the inhibition of autophagy by mTORC1 in these CDDP-resistant cells sensitizes them to metabolic inhibitors in vitro and in vivo. The authors proposed the mTOR status and the phosphorylation of ULK1S757 by mTORC1 as biomarkers for these metabolically vulnerable tumours.

Although not all of the aspects of the story are new, the manuscript reads well, and the different conclusions are clearly supported by the data exposed in the manuscript. This work fits with the scope of Nature communications and I have few comments and questions.

The specific comments are presented below, with no particular order of relevance:

- Did the authors investigate the mutation(s) which lead to the CDDP-resistance of these cells?
- The authors should include (maybe in supplementary) the histograms that would illustrate the subG2 experiments presented.
- As described in the introduction, mTOR forms two structurally and functionally distinct complexes and AZD8055 inhibits both of them. However, it seems that the data point toward mTORC1 as the authors focused on the phosphorylation status of p70 and 4EBP1. Could similar results be achieved using rapamycin or a knock-down of Raptor?

In any case, some ambiguities should be lifted in the text. When mTOR is cited, for instance lines 147/148: "displayed strongly elevated mTOR signaling, characterized by phosphorylation of mTOR target sites (p70S6KT389, 4E-BP1T37/46 and ULK1S757)", the authors should mention mTORC1

instead of mTOR as these downstream targets are specifically phosphorylated by mTORC1.

- The authors should include the total protein when including western blot. Indeed, the total level of the target protein allow to determine the phosphorylated fraction relative to the total fraction which is particularly informative to compare the impact of different treatments. This would be particularly important when the target is not phosphorylated in all the conditions of the panel.
- To complete the figure 1 and confirm the apoptotic phenotype observed with 2DG and DCA, it would be informative to include a western blot with pro-apoptotic markers.
- Figure 2a, H460 parental does not exhibit any activation of mTORC1. It would be expected that H460 parental cells exhibit high mTORC1 signalling and low AMPK. Could the authors comment this point? Is it a problem of membrane exposition?
- There is a mislabelling in Figure 5, two figure d and no figure e.
- Figure 5a and 5b: H460 parental cells reacted to Met/Phen with an activation of autophagy which is in line with AMPK activation and the subsequent inhibition of mTORC1. Surprisingly, Met and Phen treatments failed to activate AMPK in the parental cells. However, in figure 5d, AMPK is indeed activated by Met or Phen in both parental and resistant cells with or without co-treatment with AZD8055. Is it due to the knock-down of ATG7?
- Figure 7a, the blue line is not labelled for the CDDP treated panel.
- To help the readers in the analysis of the IHC analysis figure 7, a quantification of the labelling of the different markers should be added.

Reviewer #1 (Remarks to the Author):

The paper by Gremke et.al. reports that cancer cells with acquired drug resistance have enhanced mTOR signaling and that this leads to enhanced sensitivity to metabolic inhibitors due to the mTOR inhibiting autophagy. The authors use a variety of drug resistance models including models where resistance was acquired in vivo and use pharmacological and genetic methods to show that the effects are mediated through mTOR and through autophagy.

For the most part I found the data presented here to be convincing and I think that the authors conclusions are sound. My concerns are as follows.

We thank the reviewer for appreciating the quality of our manuscript and for providing constructive advice to improve the manuscript further.

1. Although I think the conclusions are correct, I am not so sure that this is a significant conceptual advance. As the authors note, mTOR is known to be a key driver of cancer drug resistance so it's not surprising that their resistant cells have upregulated mTOR signaling. mTOR is also well known to be the principle negative regulator of autophagy and autophagy is known to be a key determinant of cancer cell survival and autophagy inhibition is well known as a way to sensitize to anti-cancer treatments and metabolic dysfunction including block of glycolysis and with agents like metformin. So I think all these conclusions are what would be predicted. This is my main issue about the paper. If the authors can explain why I'm wrong, and what there is that is really new here, I'd perhaps reconsider.

The strength of our manuscript is to demonstrate in a preclinical model system that cancer drug resistance can generate a druggable metabolic vulnerability and to work out the underlying mechanisms. We agree with the reviewer that individual mechanistic aspects have been described before. However, these aspects have been described in different cell types and never before brought together and tested as a concept in a single model of cancer drug resistance, such as the CDDP-resistant lung cancer cells or PI3Ki-resistant breast cancer cells. Our study is therefore the first that synthesizes the individual findings into a unifying therapeutic strategy and validates this approach in a preclinical model, ready to be tested in a clinical setting. In addition, our results provide a possible explanation why clinical trials with many metabolic drugs have yielded controversial and often disappointing results and suggest consideration of mTOR status as a biomarker for patient stratification in future trials.

In other words, our manuscript provides the first preclinical proof-of-concept evidence for “overcoming mTOR-mediated cancer drug resistance with metabolic inhibitors” as a feasible therapeutic approach.

2. The conclusion that these effects are due to autophagy is primarily based on inactivation of ATG7. However ATG7 is also involved in other related processes like LAP or LANDO (and completely independent processes such as p53 regulation). Confirmation that similar results are obtained by targeting other genes that are more specific for autophagy such as FIP200 or ATG14 would strengthen the conclusion that this is really due to autophagy.

We thank the reviewer for pointing this out and providing helpful suggestions to better support the specific role of autophagy. As already outlined in our response to the editor's summary and recommended by the reviewer, we have added new experiments for knock-down and knock-out of FIP200 and ATG14, which are both considered specific for autophagy but not LAP or LANDO (Martinez et al., 2011; Martinez et al., 2015; Heckmann & Green, J Cell Sci 20019). In addition, we have included as a control a knock-down of RUBCN (Rubicon), which is involved in LAP but not classic autophagy (Martinez et al., 2015; Heckmann & Green, J Cell Sci 20019). Clonogenic growth assays demonstrate that abolishing FIP200 and ATG14 induces a similar degree of vulnerability to 2DG, DCA and Metformin as seen in drug-resistant H460^{res} cells, while RUBCN knock-down has no effect. We have added these new data as Supplementary Figure 3.

An essential role for p53 in the metabolic vulnerability is excluded by our data that demonstrate metabolic vulnerability of CDDP-resistant variants of p53-mutant H1975 (Fig. 1e-g) and p53-deficient murine lung cancer cells (Fig. 1h, i). In addition, we have previously described H460 cells with p53-knockout (Wanzel et al., Nat Chem Biol 2016). When these

H460-p53KO cells were made CDDP-resistant they showed the same hypersensitivity to 2DG/DCA as the H460^{res} cells in our study (data not shown).

Reviewer #2 (Remarks to the Author):

In the present MS, Gremke et al. identify a metabolic liability of cancer cells with acquired resistance to DNA damaging agents. They show that cultured human and mouse cancer cell lines evolve resistance to DNA damaging agents presumably by the activation of mTORC1 and elevating FancD2 levels. Interestingly, this mechanism of resistance sensitizes cancer cells to inhibitors of glycolysis. They further show that autophagy, a process blocked by mTORC1, is the prosurvival cellular function essential to sustain cancer cell viability in the presence of these inhibitors. The conclusions are sustained with solid data, based on several cancer cell lines and complemented with some experiments in mice, which are consistent with the results obtained in vitro. Certain aspects of the MS have been shown previously, albeit altogether, the message of the MS has provocative implications that would fit the expectations of readers of Nature Communications. Nevertheless, the MS may benefit from additional experimental support to some of the conclusions and implications.

We thank the reviewer for appreciating the quality of our study and the helpful suggestions for improvements.

-A premise of the MS is that 1- High PI3K/mTORC1 activity is a common mechanism of resistance to DNA damaging agents; and 2- Elevated FancD2 levels is a well-established mechanism to limit the efficacy of the agents. While there is published work to support for the first premise of the paper, acquired resistance to chemotherapeutic agents cannot be presented in such a simplistic manner.

We absolutely agree with the reviewer that resistance to chemotherapy is far from simplistic and - unlike resistance to some targeted drugs - usually cannot be explained by single gene alterations. As such, we did not mean to imply that CDDP resistance can be explained by increased FancD2 levels alone. We have toned down respective statements accordingly by writing that “hyperactive mTOR signaling is known to contribute to CDDP resistance, for example by upregulating FancD2”.

-The striking essentiality of autophagy for cancer cell viability in the context of glycolytic inhibitors is well established. So is the prosurvival effect of autophagy upon exposure to DNA damaging agents, a concept that seems to contradict the data presented here. In addition, increased mTORC1 activity causing the inability to survive under energetic stress has been reported many times. Several explanations have been proposed, including increased metabolic rate and energetic demand of the cell, limited nucleotide availability (which would be critical during DNA repair), and the block of autophagy. Hence, a critical experiment to support the whole model of the paper would be to ectopically express FancD2 in parental (DNA damage sensitive cells). According to the model of the MS, this should confer resistance to DNA damaging agents, and would make inhibition of PI3K-mTORC1 either with direct inhibitors (AZD8055, MLN0128, Torin1) totally innocuous during CDDP treatment. Although FancD2 is not the focus of the MS, its upregulation is an essential component of the model as it is presented.

We have previously published that FancD2 is essential for the CDDP-resistant phenotype of H460^{res} cells as knock-down of FancD2 re-sensitized these cells to CDDP (Fig. a below). As suggested by the reviewer, we have ectopically expressed FancD2 in parental H460 cells, but this did not render them CDDP-resistant, indicating that FancD2 is required but not sufficient for CDDP-resistance (Fig. b below). A possible explanation is provided by the Western blot (Fig. c below), which shows that (endogenous and ectopically expressed) FancD2 is downregulated by CDDP treatment in parental cells, while it remains stable in H460^{res} cells. The FancD2 expression mirrors mTOR signaling suggesting that sustained mTOR signaling is required to maintain high-level FancD2 expression and confer CDDP resistance. In fact, mTOR knock-down reduces FancD2 expression in CDDP-resistant cells (Fig. d below) and mTOR inhibition with AZD8055 causes FancD2 to decrease in CDDP-treated cells (manuscript Fig. 2a). We therefore believe that FancD2 expression alone is insufficient to induce CDDP resistance and requires elevated mTOR signaling to maintain increased FancD2 levels in the presence of CDDP.

However, as the reviewer already pointed out, FancD2 and the mechanisms underlying CDDP-resistance are not the focus of the manuscript. Furthermore, we never meant to imply that FancD2 is the sole cause of CDDP resistance. We have just included FancD2 in the Western blots as we know from our published work that it is upregulated by mTOR in H460^{res} cells and contributing to the resistant phenotype.

Given that FancD2 and CDDP resistance are not the main focus, we also present in Fig. 6 data on targeted therapy resistance in breast cancer which validates our model in a system completely independent of DNA damage and DNA repair. The common denominator of the drug resistance phenotype in our models is mTOR - not FancD2. We therefore feel that

including more data on FancD2 would detract from the main message and have therefore opted to not include additional FancD2 data into the manuscript.

-Besides the FancD2 connection, the lack of effect of inhibition of mTORC1 in cancer cell lines (Figure 2d and other panels) is rather surprising, more so considering the strong upregulation of the pathway in resistant cells. Acute response to mTORC1 inhibition (cell cycle arrest) should complement the end-point readouts presented. Additionally, readouts of DNA damage, such as γ -H2AX in parental and resistant cells with and without pharmacological inhibition of mTORC1 would complement the data and support the model.

mTOR inhibition certainly affected the proliferation of both parental and resistant H460 cells. While the response to Rapamycin/Everolimus was modest, dual mTORC1/mTORC2-inhibition with AZD8055 expectedly exerted stronger effects. We have included cell titer assays and proliferation curves of H460^{par} and H460^{res} cells treated with Rapamycin, Everolimus and AZD8055 as Supplementary Fig. 1.

However, as also mentioned by the reviewer, the response of H460 cells to mTOR inhibition is cytostatic and is therefore not noticed in apoptosis assays (Fig. 2e, 2g, 4g, 6d), autophagy assays (Fig. 3a-c, 5a, 6c) or Western blots (Fig. 2a, 5b, 5e). In clonogenic growth assays, where the cytostatic effect of mTOR inhibitors could have been seen, mTOR inhibitors were not applied continuously. Instead, mTOR inhibitor treatment started three days prior to the 2DG/DCA/Metformin/CDDP treatment and was discontinued after a few days of co-treatment upon the first medium change. When the plates were fixed and stained approximately 10 days later, the effects of a transient cell cycle arrest were no longer evident (Fig. 2d, 4h). While we have mentioned this treatment regime in our figure legends, it might not have been sufficiently clear that mTOR inhibitors were applied only transiently. We have removed details from the figure legends and added this information to the methods under the section “clonogenic growth assays”.

Most importantly and different from what might be expected at first sight, H460^{res} cells with elevated mTOR activity did not respond better to mTOR inhibition than parental cells. In fact, H460^{res} cells were even slightly less responsive to mTOR inhibition by Rapamycin or Everolimus (Supplementary Fig. 1). How can this be explained? It is known that mostly tumor cells with constitutive mTOR upregulation due to TSC mutations are highly susceptible to mTOR inhibition, as mTOR in this situation is a key driver of the transformed phenotype and renders such tumor cells addicted to the mTOR oncogene. Parental H460 cells are already highly tumorigenic in the absence of strong mTOR signaling and not mTOR-addicted. If these cells upregulate mTOR signaling as a resistance mechanism to better survive CDDP treatment, this does not imply that they become mTOR-addicted. Thus, mTOR inhibition in chemoresistant cells might help to restore sensitivity to chemotherapy (as shown in Fig. 2d and Supplementary Fig. 1), but does not necessarily suffice to eliminate resistant tumor cells in the absence of chemotherapy. In light of this, targeting mTOR-induced vulnerabilities as described in our manuscript would be a promising alternative approach. We have added a statement to the discussion.

-Is inhibition of autophagy a sensitizer also for DNA damaging agents in the systems utilized in the MS?

While knockdown or knockout of ATG7, FIP200, ATG14 sensitized to 2DG/DCA/metformin, we did not observe any impact on CDDP sensitivity. This is now shown in the new Supplementary Fig. 3 and mentioned in the results section.

-The *in vivo* experimental system used in Figure 1g is not exploited in Figure 7. Can the authors explain why?

We wanted to conduct the preclinical mouse study using human lung tumor cells. Lung tumors from genetically engineered mice have a mutational burden more than 100-fold lower than that of human disease (McFadden et al., PNAS 2016). As genomic instability is a key factor driving therapy resistance, this limits their translational value for resistance studies.

-Pharmacological inhibition of mTOR and genetic activation of mTOR complex 1 are presented. To support the notion that mTORC1 (and not mTORC2) is responsible for the resistance to DNA damage and for the sensitivity to 2DG and DCA, the effect of rapalogs should also be tested in some of the experiments presented, such as the assay in Figure 2d.

We thank the reviewer for suggesting this very informative experiment. We have added the requested clonogenic growth assay for the mTORC1-selective inhibitors Rapamycin and Everolimus as Supplementary Fig. 1. Rapamycin and the rapalog Everolimus rescued the metabolic vulnerability of H460^{res} to the same extent as the dual mTORC1/2 inhibitor AZD8055. To further confirm this genetically, we also knocked-down in H460^{res} cells Raptor and Rictor as specific subunits of mTORC1 and mTORC2, respectively. Only knockdown of Raptor rescued the metabolic vulnerability. Together these new experimental data allow the conclusion that the metabolic vulnerability is specifically caused by mTORC1.

Reviewer #3 (Remarks to the Author):

The manuscript by Gremke et al. provides a mechanism in which mTOR simultaneously promotes cancer cell resistance to chemotherapy and inhibits autophagy which is a necessary process for tumour cells to cope with metabolic perturbations induced by chemotherapies. This work demonstrates that cisplatin-resistant cells exhibit metabolic vulnerabilities that are mediated by mTORC1. Indeed, the inhibition of autophagy by mTORC1 in these CDDP-resistant cells sensitizes them to metabolic inhibitors *in vitro* and *in vivo*. The authors proposed the mTOR status and the phosphorylation of ULK1S757 by mTORC1 as biomarkers for these metabolically vulnerable tumours.

Although not all of the aspects of the story are new, the manuscript reads well, and the different conclusions are clearly supported by the data exposed in the manuscript. This work fits with the scope of Nature communications and I have few comments and questions.

We also thank this reviewer for the favorable evaluation of our manuscript.

The specific comments are presented below, with no particular order of relevance:

- Did the authors investigated the mutation(s) which lead to the CDDP-resistance of these cells?

We excluded mutations in TSC1 and TSC2 as a possible obvious reason for mTOR upregulation by sequencing. However, we did not look further into the genetic causes of CDDP-resistance, because the H460^{res} cells gradually lost the resistant phenotype over several months in culture when not repeatedly exposed to CDDP. We therefore believe that the CDDP resistance is more likely a regulatory or epigenetic phenomenon rather than a genetically fixed phenotype caused by mutations.

- The authors should include (maybe in supplementary) the histograms that would illustrate the subG1 experiments presented.

We have included exemplary histograms as new Supplementary Fig. 2.

- As described in the introduction, mTOR forms two structurally and functionally distinct complexes and AZD8055 inhibits both of them. However, it seems that the data point toward mTORC1 as the authors focused on the phosphorylation status of p70 and 4EBP1. Could similar results be achieved using Rapamycin or a knock-down of Raptor? In any case, some ambiguities should be lifted in the text. When mTOR is cited, for instance lines 147/148: "displayed strongly elevated mTOR signaling, characterized by phosphorylation of mTOR target sites (p70S6KT389, 4E-BP1T37/46 and ULK1S757)", the authors should mention mTORC1 instead of mTOR as these downstream targets are specifically phosphorylated by mTORC1.

As also mentioned in the response to the other reviewers, we have included further experiments with Rapamycin and Everolimus as well as with knock-down of Raptor and Rictor, which confirm that the metabolic vulnerability is driven by mTORC1. These data are included as Supplementary Fig. 1. In addition, we have replaced mTOR by mTORC1 at multiple places throughout the manuscript wherever the observed effect can be specifically attributed to mTORC1.

- The authors should include the total protein when including western blot. Indeed, the total level of the target protein allow to determine the phosphorylated fraction relative to the total fraction which is particularly informative to compare the impact of different treatments. This would be particularly important when the target is not phosphorylated in all the conditions of the panel.

Originally, we had not shown the total protein levels to save space. As requested, we have added blots for total proteins to all the figures. Of note, we were not able to reliably detect total ULK1 with the available commercial antibody in many cell lines, so that for ULK1 only phospho-ULK1 is shown.

- To complete the figure 1 and confirm the apoptotic phenotype observed with 2DG and DCA, it would be informative to include a western blot with pro-apoptotic markers.

We have added a Western blot for cleaved Parp as new Fig. 1b.

- Figure 2a, H460 parental does not exhibit any activation of mTORC1. It would be expected that H460 parental cells exhibit high mTORC1 signalling and low AMPK. Could the authors comment this point? Is it a problem of membrane exposition?

Parental H460 cells always show only low levels of mTORC1 signaling. Because of the strongly elevated signals for mTORC1 targets in H460^{res} cells we had to limit the exposure, so that this low level is not seen in Fig. 2a. However, in Fig. 2j it is seen that there is indeed a low degree of 4E-BP1 and ULK1 phosphorylation also in parental cells.

- There is a mislabelling in Figure 5, two figure d and no figure e.

We have corrected this mistake.

- Figure 5a and 5b: H460 parental cells reacted to Met/Phen with an activation of autophagy which is in line with AMPK activation and the subsequent inhibition of mTORC1. Surprisingly, Met and Phen treatments failed to activate AMPK in the parental cells. However, in figure 5d, AMPK is indeed activated by Met or Phen in both parental and resistant cells with or without co-treatment with AZD8055. Is it due to the knock-down of ATG7?

One might indeed expect to see AMPK activation in Met/Phen-treated parental cells. We believe that this expected increase in phospho-AMPK is not evident in the Western blot as it is immediately counteracted by induction of autophagy. In support of this, AMPK activation is clearly seen when autophagy is blocked (H460^{par} ATG7^{-/-} cells in Fig. 5d).

- Figure 7a, the blue line is not labelled for the CDDP treated panel.
We have corrected this.

- To help the readers in the analysis of the IHC analysis figure 7, a quantification of the labelling of the different markers should be added.

We have added a quantification for all markers and treatments as Supplementary Fig. 6 Staining intensity of 10 randomly chosen pulmonary tumor nodules was automatically quantified using the Aperio image scope positive pixel count algorithm reporting the results as the positivity index (ratio of positive and total pixels). Differences between H460^{par} and H460^{res} tumors were tested for statistical significance using two-way ANOVA and multiple comparisons testing. We left the most important (manual) quantification of apoptotic (cleaved caspase-3 positive) cells in the main figure as 7d.

Reviewers' Comments:

Reviewer #1:

Remarks to the Author:

The authors have addressed my previous concern about the possibility that they may be seeing an autophagy-independent effect very well with their new experiments targeting different genes. I am therefore confident that the authors' conclusions are well supported by their data.

However, I am still not entirely convinced about the novelty and overall significance of the work since I believe that bringing all these ideas together in a single model is a fairly incremental advance. That is however more of an editorial decision than one for a reviewer.

Reviewer #2:

Remarks to the Author:

In this revised version with the accompanying rebuttal letter, the authors have addressed my points and strengthened the value of an interesting work.

Reviewer #3:

Remarks to the Author:

In this revised version, the authors have addressed all my concerns and criticisms. I have no further questions or comments. I now fully support publication.